# A unifying mechanism for the biogenesis of membrane proteins co-operatively integrated by the Sec and Tat pathways

Fiona J Tooke[1†], Marion Babot[1†], Govind Chandra[2], Grant Buchanan[1], Tracy Palmer[1]*

[1]Division of Molecular Microbiology, School of Life Sciences, University of Dundee, Dundee, United Kingdom; [2]Department of Molecular Microbiology, John Innes Centre, Norwich, United Kingdom

**Abstract** The majority of multi-spanning membrane proteins are co-translationally inserted into the bilayer by the Sec pathway. An important subset of membrane proteins have globular, cofactor-containing extracytoplasmic domains requiring the dual action of the co-translational Sec and post-translational Tat pathways for integration. Here, we identify further unexplored families of membrane proteins that are dual Sec-Tat-targeted. We establish that a predicted heme-molybdenum cofactor-containing protein, and a complex polyferredoxin, each require the concerted action of two translocases for their assembly. We determine that the mechanism of handover from Sec to Tat pathway requires the relatively low hydrophobicity of the Tat-dependent transmembrane domain. This, coupled with the presence of C-terminal positive charges, results in abortive insertion of this transmembrane domain by the Sec pathway and its subsequent release at the cytoplasmic side of the membrane. Together, our data points to a simple unifying mechanism governing the assembly of dual targeted membrane proteins.

*For correspondence: t.palmer@dundee.ac.uk

†These authors contributed equally to this work

Competing interests: The authors declare that no competing interests exist.

## Introduction

Prokaryotic cytoplasmic membrane proteins represent 20–30% of the proteome (*Wallin and von Heijne, 1998*; *Krogh et al., 2001*) and they fulfil a wide variety of critical functions in the cell including respiration, photosynthesis, and ion transport, allowing this membrane to act as a tightly controlled barrier between the cytoplasm and the extracellular environment. Cytoplasmic integral membrane proteins adopt $\alpha$-helical topologies, and in bacteria are inserted via the action of at least one of three protein translocation machineries - the Sec machinery, the YidC insertase and the Tat pathway (see [*Collinson et al., 2015*] for a recent review).

The SecYEG translocon is the major route by which multi-spanning membrane proteins are integrated into the membrane. The insertion of transmembrane domains of polytopic proteins occurs co-translationally following targeting of the translating ribosome to the Sec machinery through the action of signal recognition particle (SRP) (*Ulbrandt et al., 1997*). YidC is positioned close to the lateral gate of SecY and interacts with nascent transmembrane domains to facilitate their integration into the membrane (*Scotti et al., 2000*; *Urbanus et al., 2001*; *Sachelaru et al., 2015*). YidC can also act independently of the Sec system to integrate small (usually mono- or bitopic) membrane proteins directly into the bilayer (*Dalbey et al., 2014*; *Samuelson et al., 2000*). The final topology adopted by a polytopic membrane protein depends upon a number of intrinsic and extrinsic factors including the hydrophobicity of membrane-spanning regions, the number and location of positively-charged amino acids and the composition of the lipid bilayer (*White and von Heijne, 2008a*; *Cymer et al., 2015*; *Bogdanov et al., 2014*).

The Tat system is a post-translational protein transport pathway that operates independently of the Sec and YidC machineries to transport folded proteins across the cytoplasmic membrane (reviewed in *Berks, 2015*; *Kudva et al., 2013*). Proteins are targeted to the Tat machinery by N-terminal signal sequences containing a highly conserved pair of arginine residues that are usually critical for efficient recognition of substrates (*Stanley et al., 2000*). A subset of Tat substrate proteins contain non-covalently bound prosthetic groups such as metal-sulphur clusters or nucleotide-based cofactors, many of which play important roles in respiratory and photosynthetic metabolism (*Palmer and Berks, 2012*). Some Tat substrates are also integral membrane proteins. In bacteria Tat-dependent integral membrane proteins generally fall into two classes – those that are N-terminally anchored in the bilayer by a non-cleaved signal sequence, such as the Rieske iron-sulfur proteins for example of *Paracoccus* or *Legionella* (*Bachmann et al., 2006*; *De Buck et al., 2007*) or the TtrA subunit of *Salmonella* tetrathionate reductase (*James et al., 2013*) and those that have a single transmembrane helix at their C-termini such as the small subunits of hydrogenases and formate dehydrogenases (*Jormakka et al., 2002*; *Hatzixanthis et al., 2003*).

Recent studies have indicated that the Rieske proteins of actinobacteria are highly unusual Tat substrates (*Keller et al., 2012*; *Hopkins et al., 2014*). Rieske proteins are essential membrane-bound components of cytochrome $bc_1$ and $b_6f$ complexes that coordinate an iron-sulfur (FeS) cluster involved in electron transfer from quinones to cytochromes $c_1/f$ (for reviews see [*Cooley, 2013*; *Baniulis et al., 2008*]). The actinobacterial proteins have three transmembrane domains (TMDs) preceding the Rieske FeS domain, unlike most other Rieske proteins which contain only one TMD. Inspection of actinobacterial Rieske sequences indicates the presence of a predicted twin-arginine motif between TMDs 2 and 3, suggesting the possibility that the concerted action of more than one translocase may be required for correct assembly. Indeed it was shown that the first two TMDs of the *Streptomyces coelicolor* Rieske protein, Sco2149, are inserted by the Sec machinery, probably in a co-translational manner, whereas the insertion of TMD3 is dependent on the Tat pathway (*Keller et al., 2012*), providing the first example of these two machineries operating together to assemble a single protein.

These findings raise a number of pertinent questions about the mechanisms by which these translocases are co-ordinated to ensure that the Sec system does not integrate TMD3 but releases the polypeptide to allow folding of the globular domain, and the subsequent recognition of a membrane-tethered substrate by the Tat pathway. It also raises the question whether actinobacterial Rieske proteins represent an oddity of nature, or whether there are further examples of dual Sec/Tat-targeted membrane proteins to be discovered. Here we have addressed both of these major aspects and show that in addition to Rieske there are at least two further conserved families of dual targeted membrane proteins across bacteria and archaea that each have 5 TMDs. A further family of proteins related to the *Bacillus subtilis* Tat substrate YkuE (*Monteferrante et al., 2012*) and predicted to have 4TMDs was also identified. A detailed dissection of the features of the transmembrane regions of *S. coelicolor* Rieske reveals that the relatively low hydrophobicity of TMD3 coupled with the location of positively charged amino acid residues orchestrate the release of the polypeptide by the Sec pathway. Importantly, we demonstrate that these features are also present across all identified families of these dual-targeted membrane proteins indicating that there is unifying mechanism for their biogenesis.

## Results

### Fusion proteins for the analysis of Sco2149 membrane assembly

Previous work has shown that the *S. coelicolor* Rieske protein, Sco2149, has three transmembrane domains that require the combined action of two distinct protein translocases, Sec and Tat, for complete assembly into the membrane (*Keller et al., 2012*; *Hopkins et al., 2014*). However the mechanism by which these two translocases are coordinated is unknown, although TMD and globular domain swapping experiments indicated that the information required to coordinate this process does not reside within the first two TMDs or the cofactor binding domain (*Keller et al., 2012*).

To assess the mechanism of TMD insertion we used constructs where the cofactor-containing FeS domain was genetically removed from Sco2149 and replaced with the mature region of two different reporter proteins – that of the *E. coli* Tat substrate AmiA (*Ize et al., 2003*) to report on interaction

of Sco2149 with the Tat pathway, or of the Sec substrate $\beta$-lactamase (Bla, which is compatible for export with either the Sec or Tat pathways depending on the nature of the targeting sequence [*Stanley et al., 2002*]) (*Figure 1A*, *Figure 1—figure supplement 1*). These constructs were produced from the medium copy number vector pSU-PROM (which specifies kanamycin resistance [*Jack et al., 2004*]) under control of the constitutive *tatA* promoter (*Jack et al., 2001*).

AmiA and its homologue AmiC are periplasmic Tat substrates that remodel the peptidoglycan, and in their absence *E. coli* is sensitive to growth in the presence of SDS (*Ize et al., 2003*; *Bernhardt and de Boer, 2003*) (*Figure 1C*; top panel). As expected, when either plasmid-encoded native AmiA or the Sco2149$_{TMD}$-AmiA fusion was produced in the *tat*$^+$ strain lacking chromosomally encoded periplasmic AmiA and AmiC (MCDSSAC), growth on SDS was restored (*Figure 1C*, middle two panels). The export of AmiA from both of these constructs was absolutely dependent on the Tat pathway as no growth on SDS was conferred in the *tat*$^-$ strain (MCDSSAC △*tat*). Previously it has been reported that a twin lysine substitution of the twin arginine motif of Sco2149 was sufficient to prevent Tat-dependent export of AmiA when produced at lower levels from the pSU18 plasmid (*Keller et al., 2012*). However, when expressed from the pSU-PROM vector, a low level of export by the Tat pathway could still be observed for the Sco2149-AmiA construct harbouring this substitution (*Figure 1—figure supplement 2*). It has been noted previously that Tat-dependent export of some very sensitive plasmid-borne reporter proteins can be detected following twin lysine substitution of the twin arginines (*Ize et al., 2002*; *Kreutzenbeck et al., 2007*), indicating that twin lysines can still trigger Tat-dependent export but with a greatly reduced efficiency. However, less conservative substitutions of the twin arginine motif to twin alanine or to alanine-aspartate were not permissive for Tat transport (*Figure 1C*; *Figure 1—figure supplement 2*).

The membrane insertion of Sco2149 was further investigated using the Bla fusion construct. When exported to the periplasmic side of the membrane Bla confers resistance to ampicillin, which can be assessed in a quantitative manner using M.I.C.Evaluator test strips. *Figure 1D* shows that the basal M.I.C. for ampicillin was evaluated at 2.5 and 1.4 µg/ml, respectively, for the *tat*$^+$ (MC4100) and *tat*$^-$ (DADE) strains harbouring the empty vector. We assign these slight differences in M.I.C. to the partially compromised cell wall in *tat* mutant strains (*Ize et al., 2003*; *Bernhardt and de Boer, 2003*). The *tat*$^+$ strain producing the Sco2149$_{TMD}$-Bla fusion protein was able to grow up to a concentration of approximately 15 µg/ml ampicillin, indicating that there was export of Bla in this strain. However, some of that export was clearly by the Sec pathway since the *tat*$^-$ strain producing Sco2149$_{TMD}$-Bla had an M.I.C. for ampicillin of 7.6 µg/ml, significantly above basal level. It has been reported that the introduction of negative charges into the n-region of a Sec signal peptide blocks Sec-dependent translocation (*Inouye et al., 1982*), and therefore substituting the twin arginines to alanine-aspartate would be expected to prevent translocation through both the Sec and Tat pathways. As shown in *Figure 1D* these substitutions reduced the MIC for ampicillin to 4.0 and 1.3 µg/ml, respectively, for *tat*$^+$ and *tat*$^-$ strain, very close to basal level. Taken together these results indicate that there is some compatibility of TMD3 of the *S. coelicolor* Rieske protein with the Sec pathway, which was not seen previously using a more qualitative assay (*Keller et al., 2012*).

## The cytoplasmic loop region of Sco2149 does not modulate interaction of TMD3 with the Sec pathway

The finding that there is some Sec-dependent translocation of the Bla portion of the Sco2149$_{TMD}$-Bla fusion in a strain lacking the Tat pathway provides a useful tool to study features of the protein that influence interaction with the Sec machinery. We therefore undertook a programme of mutagenesis on the Sco2149$_{TMD}$-Bla construct, focusing firstly on the cytoplasmic loop region between TMD2 and TMD3 as this has a number of highly conserved features across actinobacterial Rieske proteins (*Figure 1B*; *Figure 1—figure supplement 3*). In particular the loop has a highly conserved length (43 amino acids between the predicted end of TM2 and the twin arginine motif), a region of predicted α-helical structure, and a number of positions where positively or negatively charged residues are conserved, including an almost invariant glutamic acid (E127 in Sco2149) and arginine-histidine pairing (R133, H134 in Sco2149).

Initial site-directed replacement of amino acids in the loop region were undertaken and the level of resistance to ampicillin mediated by the variant Sco2149$_{TMD}$-Bla fusion protein in a *tat*$^-$ background was scored. As shown in *Table 1*, apart from the introduction of an alanine-aspartate pair to replace the twin arginines, none of the substitutions we introduced, including replacement of the

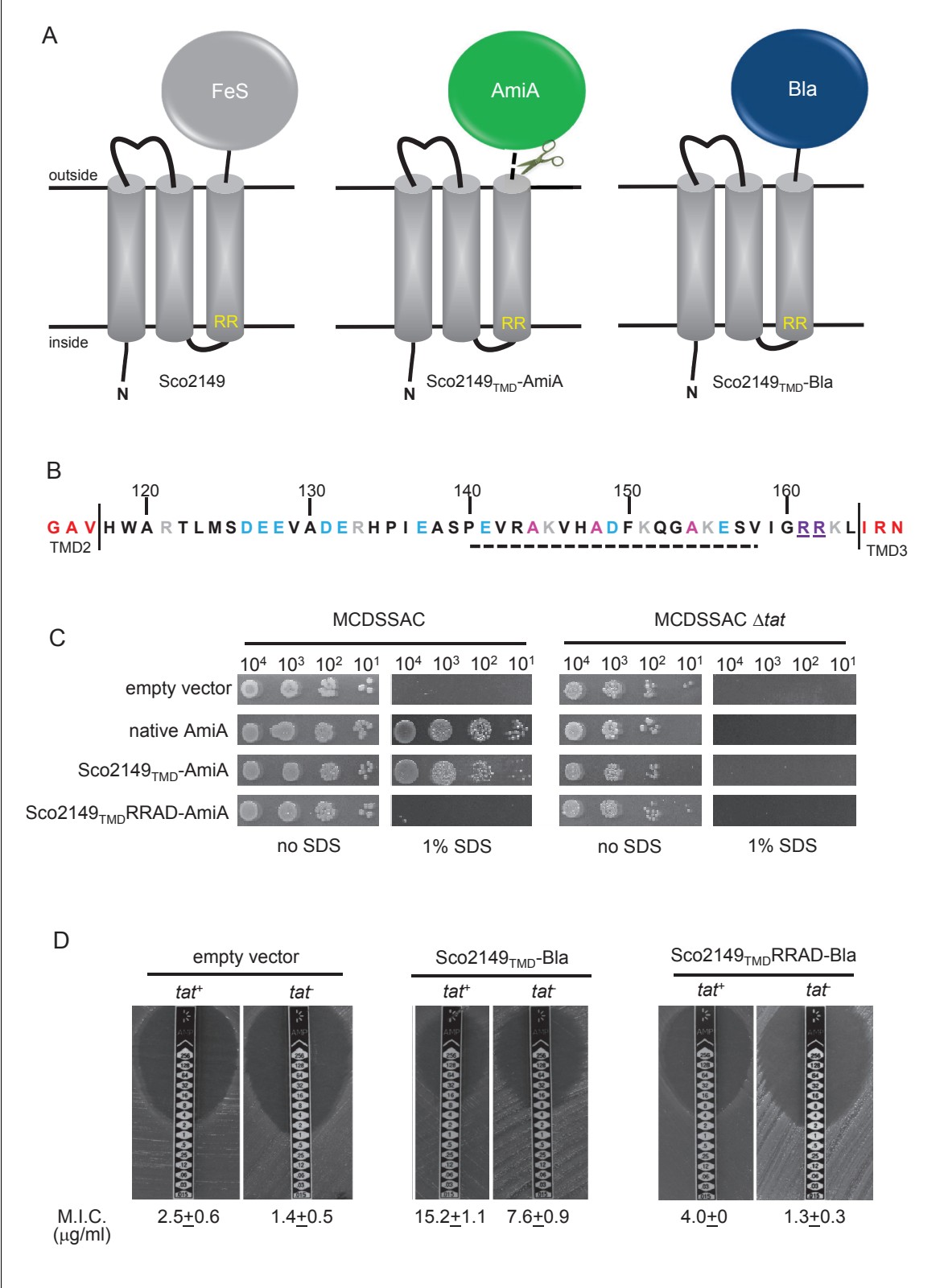

**Figure 1.** Sco2149<sub>TMD</sub>-reporter fusions to follow membrane insertion. (**A**) Cartoon representations of the *S. coelicolor* Rieske protein, Sco2149, and the Sco2149<sub>TMD</sub>-AmiA and Sco2149<sub>TMD</sub>-Bla fusions. A signal peptidase I cleavage site (indicated by scissors) was introduced between the end of TMD3 and the AmiA sequence to allow release of AmiA from the membrane (**Keller et al., 2012**). The position of the twin-arginine motif is indicated by RR. (**B**) Sequence of the Sco2149 cytoplasmic loop region between TMDs 2 and 3. Amino acids predicted to be part of TMDs 2 and 3 are shown in red. The

*Figure 1 continued on next page*

*Figure 1 continued*

twin arginines of the Tat recognition motif are given in purple underline. Predicted α-helical secondary structure is shown with a dotted line, and alanine residues within this region that were mutated to proline are shown in pink. Negatively charged amino acids in the loop region are shown in blue, positively charged ones in grey. (C) *E. coli* strain MCDSSAC (which carries chromosomal deletions in the signal peptide coding regions of *amiA* and *amiC*) or an isogenic *tatABC* mutant containing either pSU-PROM (empty vector), or pSU-PROM producing native AmiA, Sco2149$_{TMD}$-AmiA or a variant where the twin-arginines were substituted to AD (Sco2149$_{TMD}$RRAD-AmiA), were spotted, after serial dilution, on LB medium in the absence or presence of 1% SDS. The plates were incubated for 20 hr at 37°C. (D) Representative images of M.I.C.Evaluator strip tests of strains MC4100 (*tat$^+$*) and DADE (*tat$^-$*) harbouring pSU-PROM (empty vector), pSU-PROM Sco2149$_{TMD}$-Bla or pSU-PROM Sco2149$_{TMD}$RRAD-Bla are shown. The mean M.I.C ± s.d. for strains harbouring these constructs is given at the bottom of each test strip (where *n* = 4 biological replicates for each strain harbouring the empty vector, *n* = 5 biological replicates for each strain harbouring pSU-PROM Sco2149$_{TMD}$-Bla and *n* = 3 biological replicates for each strain harbouring pSU-PROM Sco2149$_{TMD}$RRAD-Bla).

The following figure supplements are available for figure 1:

**Figure supplement 1.** Sequence alignment of selected actinobacterial Rieske proteins.

**Figure supplement 2.** A twin lysine substitution of the twin arginine motif of Sco2149 still retains some interaction with the Tat pathway.

**Figure supplement 3.** Sequence alignment TMD2/TMD3 loop region for a selection of actinobacterial Rieske proteins.

**Figure supplement 4.** Effect of ≥35 residue truncations in the Sco2149 cytoplasmic loop region on the ability of Sco2149$_{TMD}$-AmiA to support growth on SDS.

highly conserved E127 or R133/H134 residues or introduction of proline residues into the predicted α-helical region, had any substantive effect on the interaction of Sco2149 with the Sec pathway. We therefore made further substitutions, for example progressively deleting clusters of negatively charged amino acids or changing them to positively charged lysines. None of these deletions or substitutions had any detectable effect on Sec translocation of the Bla fusion, even when all of the acidic residues were substituted for lysine. Moreover, insertion of three additional negative charges into the loop was also without detectable effect.

We similarly assessed translocation by Sec for a series of sliding truncations of 5, 10, 15, 20, 25, 30 and 35 residues within the loop region (summarised in *Table 2*). Again most of the truncations had little effect on translocation of Sco2149$_{TMD}$-Bla by the Sec pathway, and even truncations of 30 residues or more gave mean M.I.C.s for ampicillin similar to that seen for the non-mutated construct. These findings indicate that many of the conserved features noted in this loop region, for example the overall length, presence of a predicted α-helical region and clusters of negatively charged amino acids do not modulate interaction of Sco2149 with the Sec pathway.

We did note, however, that one of the 35 residue truncations, Δ123–157, significantly reduced integration of TMD3 by the Sec pathway (*Figure 2A,B*), whereas the other 35 residue truncation, Δ118–152, showed a slight increase in Sec translocation (c.f. M.I.C of 7.6 μg/ml ampicillin for the non-mutated construct vs 12 μg/ml for the Δ118–152 truncation). This suggested that there may be some feature of the loop region between residues 153 and 157 influencing interaction with the Sec pathway. To explore this further we made a series of additional one amino acid truncations to give Δ118–153, Δ118–154, Δ118–155 and Δ118–156 and Δ118–157 constructs. *Figure 2B* indicates that as soon as the truncation extended to amino acid 155, Sec translocation was substantially reduced (but protein production and/or stability was not, *Figure 2C*). Inspection of the sequence indicates that amino acid 155 is a lysine. Positively charged amino acids are important topology determinants in membrane proteins, and are enriched in the cytoplasmic regions of membrane proteins, the so-called 'positive inside rule' due to the energetic cost of translocating them across the membrane against the protonmotive force (*Heijne, 1986*; *Nilsson and von Heijne, 1990*). To test whether the loss of this basic residue was the reason for the very low level of periplasmic Bla activity, we introduced a positive charge further along the loop (V158K) into the full length Sco2149$_{TMD}$-Bla and the Δ118–155, Δ118–156 and Δ118–157 truncations. *Figure 2D* shows that the introduction of the V158K into the Δ118–155, Δ118–156 and Δ118–157 truncations restored the M.I.C. to a similar level

**Table 1.** Effect of amino acid substitutions, small deletions and insertions in the Sco2149 cytoplasmic loop region on the ability of Sco2149$_{TMD}$-AmiA and Sco2149$_{TMD}$-Bla to support growth on SDS or ampicillin, respectively. Note that growth on ampicillin was scored using the *tat*⁻ strain DADE and therefore assesses Sec translocation only. Y indicates growth on 1% SDS, N indicates no growth, nd – not determined. Mean M.I.C for growth on ampicillin is given in µg/ml $\pm$ one standard deviation, *n* = at least 3. *Insertion of 3 additional amino acids, DEE between E128 and V129.

| Variant | Growth on 1% SDS (Tat translocation) | Mean M.I.C. for ampicillin (Sec translocation) |
|---|---|---|
| wild type | Y | 7.6 ± 0.9 |
| R161K R162K | Y | 3.8 ± 0.5 |
| R161A R162D | N | 1.3 ± 0.3 |
| R161A R162A | N | 5.5 ± 1.0 |
| R161K R162Q | N | 3.3 ± 0.6 |
| ΔR161 ΔR162 | N | 9.3 ± 2.3 |
| R133H H134R | Y | 7.3 ± 1.2 |
| R133K H134K | Y | 7.0 ± 2.0 |
| M124L | Y | 6.0 ± 2.0 |
| M124A | Y | 6.0 ± 2.0 |
| S125L | Y | 6.7 ± 2.3 |
| S125A | Y | 8.0 ± 0.0 |
| D126L | Y | 7.3 ± 1.2 |
| D126A | Y | 7.3 ± 1.2 |
| E127L | Y | 6.0 ± 2.0 |
| E127A | Y | 8.0 ± 2.8 |
| A144P | Y | 6.7 ± 1.2 |
| A148P | Y | 8.0 ± 2.8 |
| A154P | Y | 8.0 ± 0.0 |
| Δ126–8 | nd | 8.0 ± 0.0 |
| Δ126–127 | nd | 7.5 ± 1.0 |
| Δ127–128 | nd | 8.0 ± 2.8 |
| 131–2 | nd | 7.3 ± 1.2 |
| Δ137Δ141 | nd | 4.0 ± 0.0 |
| Δ131–2Δ141 | nd | 6.0 ± 2.0 |
| Δ149Δ156 | nd | 8.0 ± 0.0 |
| Δ126–8 Δ137Δ141 | nd | 8.0 ± 0.0 |
| Δ126–8Δ131–2 | nd | 8.0 ± 2.4 |
| Δ126–8Δ131–2 Δ137Δ141 | nd | 10.4 ± 2.2 |
| Δ126–8Δ131–2 Δ137 Δ141 Δ149 Δ156 | nd | 10.0 ± 2.8 |
| Ins D129 E130 E131* | nd | 6.7 ± 1.2 |
| D131A E132A | nd | 6.0 ± 2.3 |
| E137A E141A | nd | 6.7 ± 2.3 |
| D126A E127A E128A | nd | 7.2 ± 1.1 |
| D131K E132K | nd | 7.3 ± 1.2 |
| E137K E141K | nd | 6.5 ± 1.9 |
| D126K E127K E128K | nd | 9.0 ± 2.0 |

*Table 1 continued on next page*

Table 1 continued

| Variant | Growth on 1% SDS (Tat translocation) | Mean M.I.C. for ampicillin (Sec translocation) |
|---|---|---|
| D126K E127K E128K E137K E141K | nd | 8.0 ± 0.0 |
| D126K E127K E128K D131K E132K E137K E141K | nd | 10.0 ± 2.2 |

Source data 1. Images of SDS growth tests and M.I.C.Evaluator strip tests of all strain and plasmid combinations used in *Table 1*.

seen for the full length Sco2149$_{TMD}$-Bla, establishing that positive charged residues in this loop region influence interaction of Sco2149 with Sec.

## A minimum cytoplasmic loop length is necessary for Tat recognition of Sco2149 TMD3

Since none of the conserved features in the Sco2149 cytoplasmic loop were required for modulating interaction with the Sec pathway, we next addressed whether they were required for recognition by the Tat system. A subset of the amino acid substitutions and each of the sliding truncations was introduced into the Sco2149$_{TMD}$-AmiA fusion protein and expressed in a *tat*$^+$ strain to allow Tat-dependence to be scored by testing for growth in the presence of SDS (*Tables 1* and *2*). *Table 1* shows that, apart from substitutions at the twin arginine motif, none of the other variants affected Tat-dependent export of AmiA, including the introduction of prolines within the predicted α-helical structure, or substitution of the highly conserved E127 or R133/H134. These results suggest that none of these features are required for recognition of the loop region by the Tat pathway.

Ordinarily, Tat signal peptides have free N-termini, whereas the Tat signal sequence of Sco2149 is internal and is only recognised by the Tat pathway once the first 2 TMD of the protein have been integrated by Sec. The loop truncation experiments indicated that the Tat system was still able to identify and integrate TMD3 when it was truncated by up to 30 residues. However, one of the 35 residue truncations (Sco2149$_{TMD}$-Δ123–157-AmiA) and the 40 residue truncation (Sco2149$_{TMD}$-Δ118–157-AmiA) supported no growth on SDS-containing media (*Table 2*; *Figure 1—figure supplement 4*), indicating that there is a minimum loop length requirement of approximately eight amino acids between TMD2 and the twin arginine motif is required for Tat recognition of a tethered signal peptide.

Taken together we conclude that, with the exception of the twin arginine motif, none of the conserved features of cytoplasmic loop are strictly necessary for interaction of Sco2149 with the Tat pathway or to mediate release from Sec.

## Specific physical properties of TMD3 drive its release from Sec

Hydrophobicity is the driving force for the insertion of a helix into the membrane (*White and von Heijne, 2008a*; *Hessa et al., 2005*; *von Heijne, 1997*). Analysis of transmembrane helices from polytopic proteins of known three-dimensional structure shows a general trend that the first and last TMDs are of similar hydrophobicity, and they are notably more hydrophobic than the central helices (*Hedin et al., 2010*; *Virkki et al., 2014*). An analysis of the apparent △G for the insertion of the three TMDs of selected actinobacterial Rieske proteins is shown in *Table 3*. It can be seen that the first and second TMDs have negative predicted △G$_{app}$ values and are therefore expected to be inserted as TMDs by the Sec system (*Ojemalm et al., 2013*). However, the third and final TMD is predicted to have a positive G$_{app}$ (*Table 3*). This is in contrast to the final TMD of 'standard' Sec-dependent proteins and suggests that this helix might be poorly recognised by the Sec machinery.

To probe this further we investigated the effect of increasing the hydrophobicity of TMD3. *Table 3* shows that substitution of a single leucine residue at either serine 179 or glycine 180 reduces the predicted △G$_{app}$ value for TMD3 Sec-dependent membrane insertion by at least 0.6 kcal mol$^{-1}$. Accordingly, when these single substitutions were individually introduced into the

**Table 2.** Effect of amino acid truncation in the Sco2149 cytoplasmic loop region on the ability of Sco2149$_{TMD}$-AmiA and Sco2149$_{TMD}$-Bla to support growth on SDS or ampicillin, respectively. Note that growth on ampicillin was scored using the *tat⁻* strain DADE and therefore assesses Sec translocation only. Y indicates growth on 1% SDS, N indicates no growth. Mean M.I.C for growth on ampicillin is given in μg/ml $\pm$ one standard deviation, *n* = at least 3.

| Variant | Growth on 1% SDS (Tat translocation) | Mean M.I.C. for ampicillin (Sec translocation) |
|---|---|---|
| wild type | Y | 7.6 ± 0.9 |
| 5 residue truncations | | |
| Δ118–122 | Y | 10.0 ± 2.3 |
| Δ123–127 | Y | 5.3 ± 2.3 |
| Δ128–132 | Y | 12.0 ± 0.0 |
| Δ133–137 | Y | 6.7 ± 2.3 |
| Δ138–142 | Y | 7.0 ± 2.0 |
| Δ143–147 | Y | 6.7 ± 2.3 |
| Δ148–152 | Y | 10.0 ± 3.3 |
| Δ153–157 | Y | 9.6 ± 3.6 |
| 10 residue truncations | | |
| Δ118–127 | Y | 6.0 ± 0.0 |
| Δ128–137 | Y | 9.5 ± 3.0 |
| Δ138–147 | Y | 8.0 ± 0.0 |
| Δ148–157 | Y | 9.3 ± 2.3 |
| 15 residue truncations | | |
| Δ118–132 | Y | 9.0 ± 3.8 |
| Δ123–137 | Y | 8.0 ± 0.0 |
| Δ128–142 | Y | 10.0 ± 2.3 |
| Δ133–147 | Y | 9.3 ± 2.3 |
| Δ138–152 | Y | 5.0 ± 1.4 |
| Δ143–157 | Y | 4.8 ± 1.5 |
| 20 residue truncations | | |
| Δ118–137 | Y | 7.3 ± 1.2 |
| Δ138–157 | Y | 5.5 ± 1.9 |
| 25 residue truncations | | |
| Δ118–142 | Y | 12.0 ± 0.0 |
| Δ123–147 | Y | 7.6 ± 0.9 |
| Δ128–152 | Y | 8.0 ± 0.0 |
| Δ133–157 | Y | 6.0 ± 0.0 |
| 30 residue truncations | | |
| Δ118–147 | Y | 10.0 ± 2.8 |
| Δ123–152 | Y | 13.6 ± 2.2 |
| Δ128–157 | Y | 7.3 ± 1.2 |
| ≥35 residue truncations | | |
| Δ118–152 | Y | 12.0 ± 0.0 |
| Δ123–157 | N | 3.0 ± 0.0 |
| Δ118–153 | Y | 16.0 ± 0.0 |
| Δ118–154 | Y/N | 9.3 ± 2.3 |
| Δ118–155 | N | 4.0 ± 0.0 |

*Table 2 continued on next page*

*Table 2 continued*

| Variant | Growth on 1% SDS (Tat translocation) | Mean M.I.C. for ampicillin (Sec translocation) |
|---|---|---|
| Δ118–156 | N | 4.0 ± 0.0 |
| Δ118–157 | N | 2.5 ± 0.6 |

Source data 1. Images of SDS growth tests and M.I.C.Evaluator strip tests of all strain and plasmid combinations used in *Table 2*.

Sco2149$_{TMD}$-Bla fusion in *tat$^-$* cells, a dramatic increase in M.I.C for ampicillin of up to 25 fold was observed (*Figure 3B*), almost at the upper limit of detection. Combining these substitutions (S179L, G180L), and including a third substitution (P177L) shifts the predicted $\triangle G_{app}$ value closer to that of TMD1 (*Table 1*). These substitutions also significantly increased the observed M.I.C. over the unsubstituted fusion, but did not appear to have additive effects over the single leucine substitutions. We conclude that the low hydrophobicity of TMD3 is a key driver for the release of Sco2149 from the Sec machinery.

It has long been known that Tat signal peptides frequently contain one or more positive charges in their c-regions, close to the site of signal peptidase cleavage. These charges are not required for the interaction with the Tat pathway but reduce the efficiency of interaction with Sec and have therefore been described 'Sec-avoidance' motifs (*Bogsch et al., 1997*; *Cristóbal et al., 1999*; *Blaudeck et al., 2001*). A positive charge is generally also found close to the C-terminal end of TMD3 of actinobacterial Rieske proteins (R185 in the case of Sco2149; *Figure 3A*, *Figure 1—figure supplement 1*). Substitution of R185 for alanine in the Sco2149$_{TMD}$-Bla fusion conferred an 8-fold increase in M.I.C for ampicillin, and therefore R185 also appears to act as a Sec-avoidance motif in this context. Interestingly, closer inspection of actinobacterial Rieske proteins indicates that there are a number of further non-conserved positive charges located within the C-terminal vicinity of TMD3 (*Figure 3A* underlined residues, *Figure 1—figure supplement 1* orange residues) which are not found in other Rieske proteins that only contain a single TMD (*Figure 1—figure supplement 1*). Since our original Sco2149$_{TMD}$-Bla fusion (where the Bla sequence is fused immediately after R185) lacks most of these additional charges (*Figure 3A*), we made an additional Bla fusion where the Sco2149 sequence in the fusion protein was extended to aa205, incorporating an additional four positively charged residues. It can be seen that inclusion of this additional positively charged stretch almost completely abolished transport via Sec, as the clearance zone around the M.I.C. strip was of similar size to that of the negative control (*Figure 3C*). We did, however, note that for unknown reasons there was a variable level of breakthrough growth within the zone of clearing for strain DADE producing the extended Sco2149$_{TMD}$-Bla fusion. We therefore constructed similar Bla fusions after TMD3 of the *M. tuberculosis* Rieske protein, QcrA. *Figure 3D* indicates that there is some Sec-dependent export of the Bla fusion when it is fused close to the C-terminal end ('short fusion') but that this was almost abolished when the sequence was extended to introduce the positively charged stretch ('long fusion'). Taken together, we conclude that a combination of low hydrophobicity of TMD3 coupled with the presence of several C-terminal positive charges promotes release of actinobacterial Rieske proteins from the Sec machinery.

## Bioinformatic analysis identifies further families of membrane proteins potentially dependent on both Sec and Tat pathways

We next asked whether actinobacterial Rieske proteins were the only protein family that required both Sec and Tat pathways for their integration. To this end, all proteins from prokaryotic genomes available in Genbank were analysed by both TATFind 1.4 (*Rose et al., 2002*) and TMHMM 2.0c (*Krogh et al., 2001*) programs, initially to identify proteins with a similar N-in topology as actinobacterial Rieske proteins. For each protein, both outputs were combined to identify the position of twin arginine motif, and the number of transmembrane helices present N-terminal and C-terminal to it. The final output from this search was sorted to give those proteins that had a predicted even number of TMDs prior to the twin-arginine motif and that had a predicted single TMD immediately

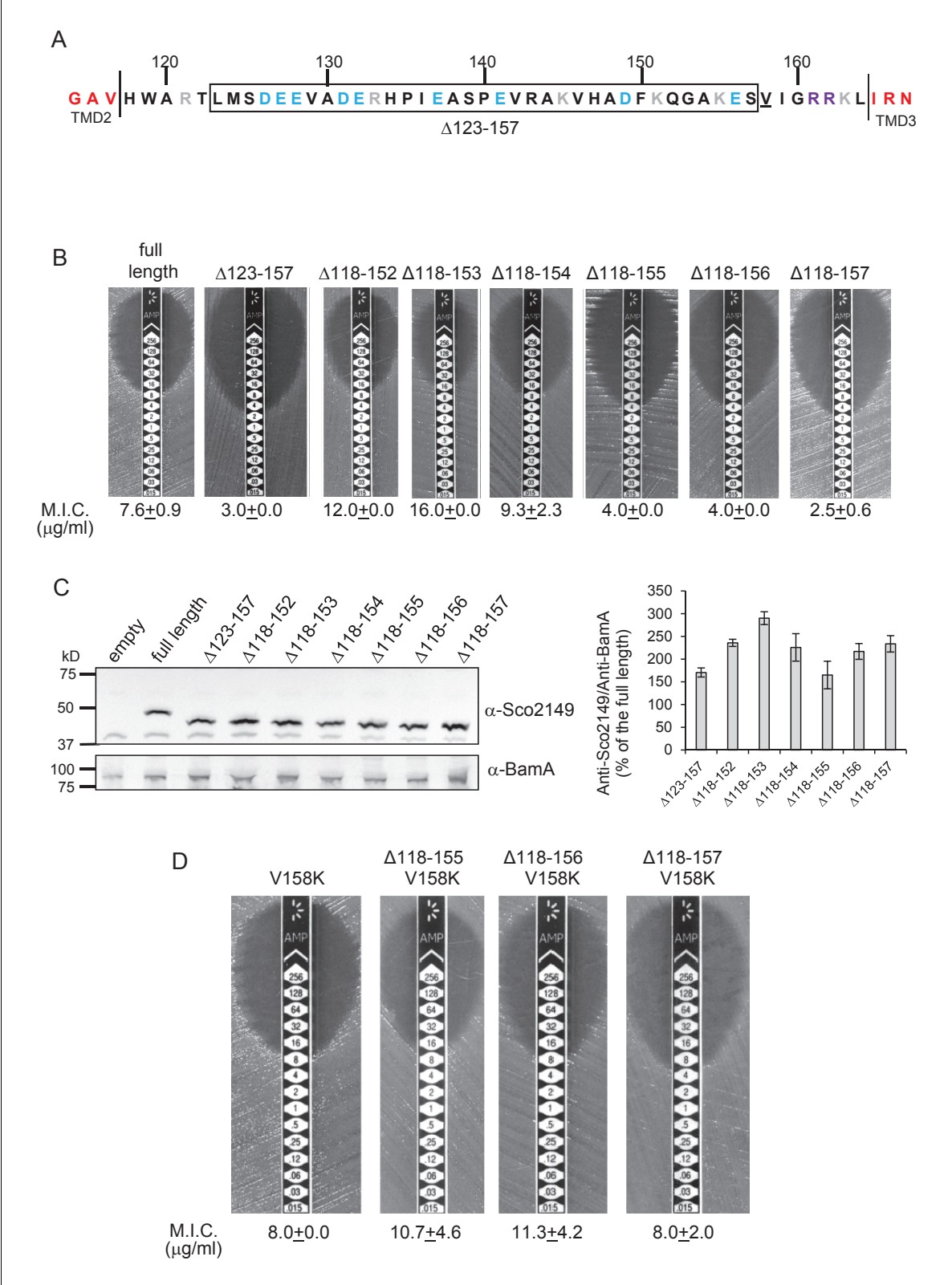

**Figure 2.** Interaction of Sco2149$_{TMD}$-Bla with the Sec pathway. (**A**) The Sco2149 cytoplasmic loop region between TMDs 2 and 3. Color-coding is as described in **Figure 1**. The extent of the 123–157 deletion is shown boxed and V158 that was substituted to K in this study is underlined. (**B**) and (**D**) Representative images of M.I.C.Evaluator strip tests of strain DADE (*tat⁻*) harbouring pSU-PROM producing the indicated variants of Sco2149$_{TMD}$-Bla. The mean M.I.C ± s.d. is given at the bottom of each test strip (where *n* = 3 biological replicates for each strain). (**C**) Membrane extracts prepared from

*Figure 2 continued on next page*

*Figure 2 continued*

the same strains used in (**B**) along with DADE harboring the empty plasmid vector as a negative control, were separated by SDS-PAGE (12% acrylamide), transferred to nitrocellulose membrane and probed with anti-Sco2149 or anti-BamA (an unrelated outer membrane protein was used as a loading control). To the right, the Sco2149-associated signal was quantified and normalised against the BamA signal for each sample. The quantification results were expressed as percentage of the normalised signal obtained for the full length fusion (which was set at 100%). The results represent mean ± s.e.m. of three biological replicates, a representative blot is shown.

The following source data is available for figure 2:

**Source data 1.** Images of Sco2149 and BamA western blots used for quantification in *Figure 2C*.
**Source data 2.** Quantification of density associated with Sco2149 and BamA signals from western blots from *Figure 2—source data 1* used to generate graph in *Figure 2C*.

following the twin-arginine motif (available as a supplementary online file at: http://www.lifesci.dundee.ac.uk/groups/tracy_palmer/docs/CombinedTATFindTMHMMoutput.docx). We subsequently manually searched this list to identify any proteins with a predicted C-terminal cofactor-binding domain.

From the output we identified a further actinobacterial Rieske homologue from *Kitasatospora setae* (KSE_30950) that is predicted to have five TMDs, with the twin arginine motif adjacent to TMD5. We also identified two further families of predicted metalloproteins that shared features of dual-inserted proteins (shown schematically in *Figure 4A*). Sco3746, also from *S. coelicolor* is predicted to have five TMDs, with a predicted molybdenum cofactor (MoCo) binding domain at the

**Table 3.** Predicted $\triangle G_{app}$ values (in kcal mol$^{-1}$) for membrane insertion of each of the three TMDs of the indicated Rieske proteins. Sequences were analysed using the $\triangle G_{app}$ prediction server (http://dgpred.cbr.su.se/) that are based on hydrophobicity scales generated from (**Hessa et al., 2005**, **2007**). This server uses the SCAMPI2/TOPCONS servers (**Tsirigos et al., 2015**, **2016**) to predict the positions of the TMDs and for *S. coelicolor* Rieske predicts TMD1 to span aa 58–80, TMD two to span aa 96–117 and TMD3 to span aa 168–187.

| Family/Species | Uniprot ID | Predicted $\triangle G_{app}$ | | | |
| | | TM1 | TM2 | TM3 | TM3/Bla fusion* |
|---|---|---|---|---|---|
| *Mycobacterium tuberculosis* | P9WH23 | −4.252 | −0.715 | 0.248 | |
| *Corynebacterium glutamicum* | Q79VE8 | −2.873 | −0.761 | 0.564 | |
| *Gordonia malaquae* | M3VAA9 | −2.569 | −0.214 | 1.291 | |
| *Corynebacterium diphtheriae* | Q6NGA2 | −2.402 | −1.089 | 0.619 | |
| *Dietzia cinnamea* | E6JC04 | −2.997 | −0.391 | 0.409 | |
| *Salinispora tropica* | A4 × 9Y7 | −2.315 | −1.205 | 1.162 | |
| *Streptomyces sp.* | D9VGG2 | −1.291 | −1.203 | 0.967 | |
| *Verrucosispora maris* | F4F1U1 | −2.248 | −1.946 | 1.162 | |
| *Stackebrandtia nassauensis* | D3Q1119 | −1.876 | −1.289 | 0.556 | |
| *Rhodococcus erythropolis* | C3JJ95 | −2.692 | −0.208 | 0.912 | |
| *Streptomyces coelicolor* | Q9 × 807 | −2.374 | −0.117 | 0.614 | 0.714 |
| *S. coelicolor S179L* | | | | −0.389 | −0.205 |
| *S. coelicolor G180L* | | | | −0.037 | 0.044 |
| *S. coelicolor S179L, G180L* | | | | −1.117 | −0.987 |
| *S. coelicolor P177L, S179L, G180L* | | | | −2.510 | −2.409 |
| *S. coelicolor R185A* | | | | 0.517 | 0.620 |

*Bla is fused to *S. coelicolor* Rieske after amino acid 185 (full sequence of all of the fusion proteins used in this study can be found in *Supplementary file 1D*).

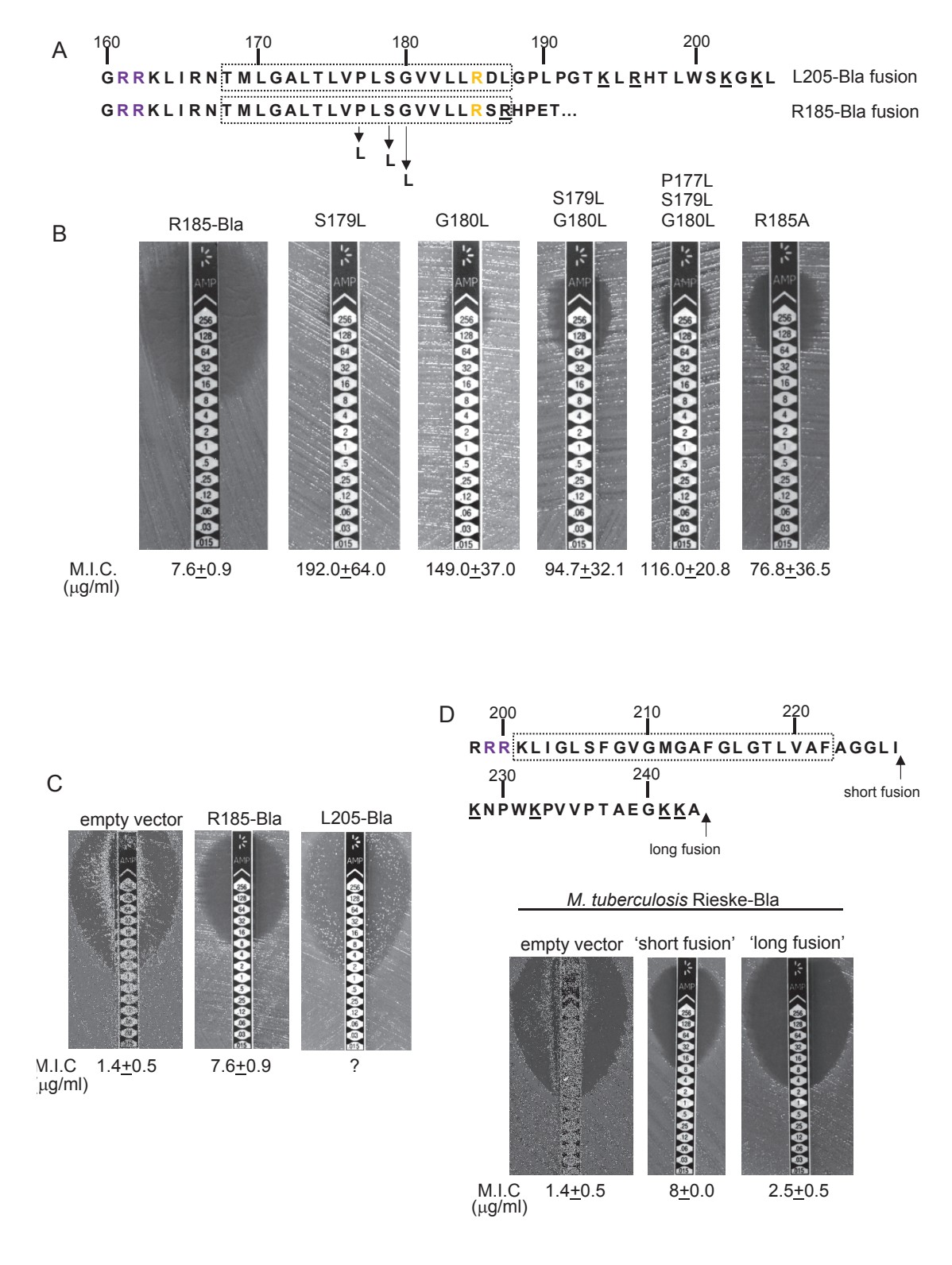

**Figure 3.** Hydrophobicity of TMD3 and C-terminal positive charges modulate interaction with the Sec pathway. (**A**) Sco2149 TMD3 and flanking sequences. *Top* shows the native Sco2149 sequence up to amino acid 205 (position of the L205-Bla fusion) and *below* the sequence of the R185 Sco2149-Bla fusion. In each case the predicted position of TMD3 was determined using the SCAMPI2/TOPCONS servers (***Tsirigos et al., 2015, 2016***) and is shown boxed. The twin arginines are shown in purple and R185 (the position after which Bla was fused in the R185 construct) is shown in

*Figure 3 continued on next page*

Figure 3 continued

yellow. Positively charged amino acids C-terminal to R185 are underlined. (**B–D**) Representative images of M.I.C.Evaluator strip tests of strain DADE (*tat*) harbouring pSU-PROM producing (**B**) the indicated variants of the R185 Sco2149$_{TMD}$-Bla fusion or (**C**) the R185 or L205 Sco2149$_{TMD}$-Bla fusions, as indicated, or (**D**) *M. tuberculosis* QcrA fused to Bla. In (**D**) the amino acid sequence around TMD3 of *M. tuberculosis* QcrA is shown, with TMD3 boxed. 'short fusion' refers to a Bla fusion after I227 and 'long fusion' to a Bla fusion after A243. In each panel the mean M.I.C ± s.d. is given at the bottom of each test strip (where *n* = 3 biological replicates for each strain).
The following source data is available for figure 3:

**Source data 1.** Images of M. I.C.Evaluator strip tests used to generate mean M.I.C. values in *Figure 3*.

C-terminus and conserved histidine residues in TMDs 2, 3 and 4 that are predicted to co-ordinate two heme *b* moieties (*Figure 4A*). The twin arginine motif, which is conserved across homologous proteins (*Figure 5*), directly precedes TMD5. Homologues of Sco3746 were identified across the actinobacteria, as well as in firmicutes, chloroflexi and euryarchaeota, and each carries a twin arginine motif directly preceding TMD5 (Examples from each phyla are shown in *Figure 5*). Protein Q1NSB0 from the delta proteobacterium *MLMS-1* is also predicted to have five TMDs and to contain seven 4Fe-4S clusters, three at the cytoplasmic side and four at the extracellular side of the membrane (*Figure 4A*; *Figure 6*). Labelling of TMD2-4 (*Figure 6*) was complicated by the observation that the iron-sulfur cluster binding regions were variably called as TMDs by some prediction programs. Again the conserved twin arginine motif directly precedes TMD5 and homologues of this protein are encoded in many prokaryotic genomes including those from the chloroflexi, nitrospirae and euryarchaeota phyla (*Figure 6*).

We subsequently modified our search to ascertain whether there might be any candidate dual-targeted proteins with an N-out topology (supplementary online file available at: http://www.lifesci.dundee.ac.uk/groups/tracy_palmer/docs/CombinedTATFindTMHMMoutput%20N-out%203.docx). From this we identified a further protein family of predicted metallophosphoesterases closely related to the *B. subtilis* Tat substrate YkuE (*Figure 4A*, *Figure 7*). *B. subtilis* YkuE has a cleavable N-terminal Tat signal peptide and lacks any TMD, and has been shown to localize to the cell wall by electrostatic interactions (*Monteferrante et al., 2012*). These longer variants of YkuE are predicted to have 4TMD and an N-out orientation, with a conserved twin arginine motif directly preceding TMD4 (*Figure 4A*, *Figure 7*). Homologues of this protein are encoded by Gram-positive and Gram-negative bacteria including those from the Firmicutes and Bacteroidetes phyla (*Figure 7*).

## Reporter proteins fused to Sco3746 or predicted polyferredoxin from *MLMS-1* are translocated by the Tat pathway

To confirm that the newly identified MoCo or polyferredoxin proteins were indeed Tat substrates, we designed constructs whereby the predicted five TMDs of Sco3746 or *MLMS-1* polyferredoxin (PFD; cloned as a synthetic gene) were fused to the reporter proteins AmiA or maltose binding protein (MBP; *Figure 4B*; exact positions of the fusions are shown in *Figures 5* and *6*). As shown in *Figure 4C*, *E. coli malE⁻* cells harboring MBP fused to these regions of either protein decolorized maltose minimal medium containing the pH indicator dye bromocresol purple. This indicates that the MBP portion of the fusion protein has been translocated to the periplasmic side of the membrane. To confirm that this translocation was dependent on the Tat pathway, the twin-arginines of the Tat recognition motif were substituted for two lysines. This conservative substitution abolished maltose fermentation (*Figure 4C*), indicating that MBP translocation was dependent on the Tat pathway. Similar findings were made using the AmiA reporter fusions. *Figure 4D* shows that, as expected, when either plasmid-encoded Sco3746$_{TMD}$-AmiA or PFD$_{TMD}$-AmiA was produced in the *tat⁺* strain lacking native AmiA/C, growth on SDS was supported. Export was dependent on the Tat pathway since growth on SDS was not supported in the *tat⁻* strain, or in the *tat⁺* strain if the twin arginine motif was substituted for twin lysine. We conclude that Sco3746 and PFD are dependent on the Tat pathway for their assembly.

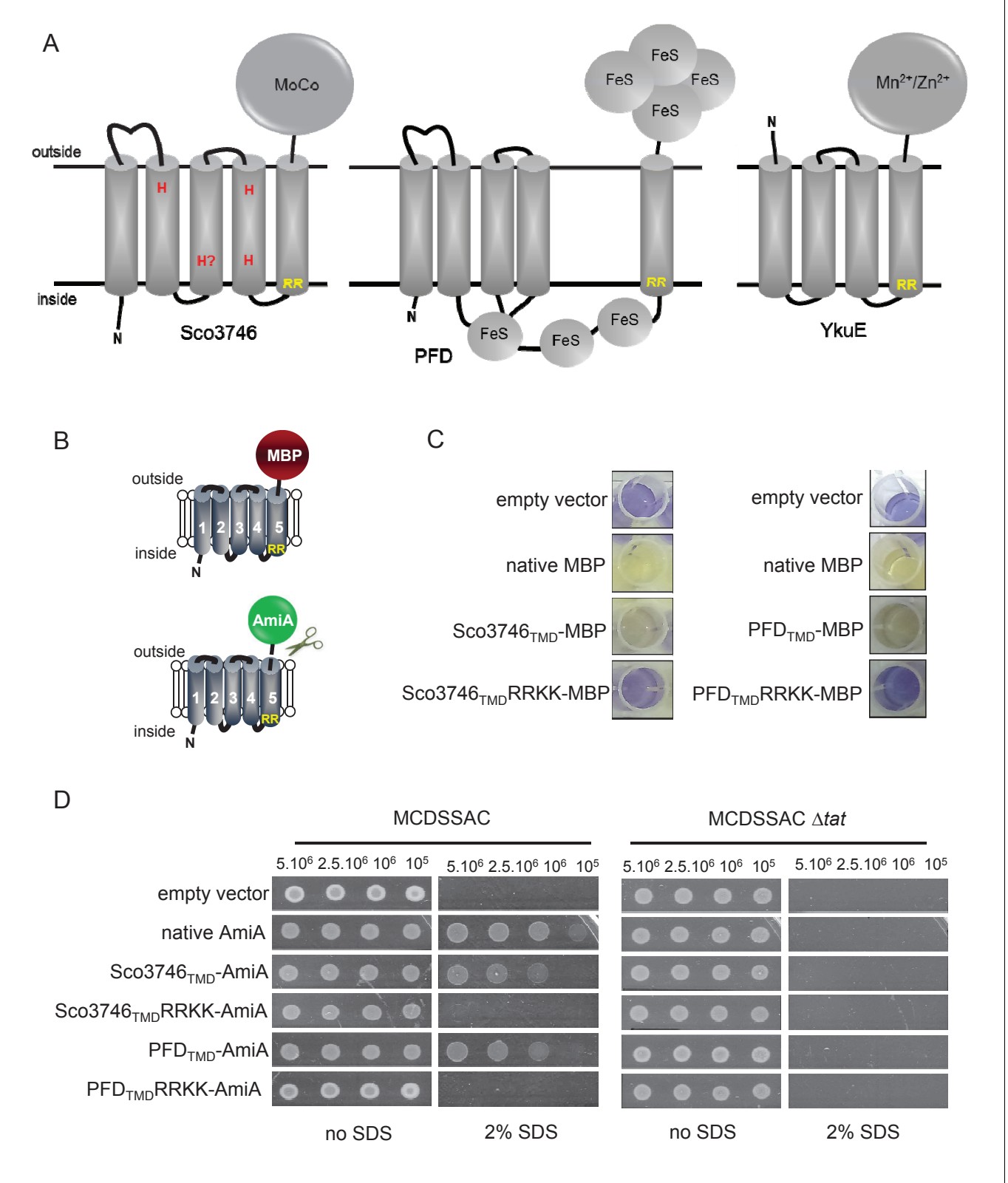

**Figure 4.** Further families of Tat-dependent polytopic membrane proteins. (A) Schematic representation of a polytopic predicted molybdenum cofactor (MoCo) binding protein (Sco3746, left) a polytopic polyferredoxin (PFD, centre) and a polytopic metallophosphoesterase of the YkuE family (right) identified bioinformatically as candidate dual-inserted membrane poteins. The twin arginines of the Tat recognition sequence are highlighted in yellow. Four histidines in the TMDs of Sco3746 and homologues that are predicted to ligate two *b* hemes are shown in red. Note that three of these histidines

*Figure 4 continued on next page*

Figure 4 continued

are conserved (**Figure 5**) but the one at the N-terminal end of TMD3 (H?) is not. (**B**) Fusions of the TMDs of Sco3746 and of Q1NSB0 (PFD) to maltose binding protein (MBP) or AmiA are shown as cartoons. As before, a signal peptidase I cleavage site (indicated by scissors) was introduced at the end of predicted TMD5 to allow release of AmiA from the membrane (**Keller et al., 2012**). (**C**) *E. coli tat*$^+$ strain HS3018-A (that lacks chromosomally-encoded MBP) harboring pSU18 (empty vector), or the same vector producing native MBP, Sco3746$_{TMD}$-MBP, PFD$_{TMD}$-MBP or the twin-arginine substituted variants Sco3746$_{TMD}$RRKK-MBP and PFD$_{TMD}$RRKK-MBP, as indicated, was cultured overnight, resuspended in minimal medium containing 1% Maltose and 0.002% Bromocresol purple. Cells were diluted either 2.5 fold (left hand panel) or 10 fold (right hand panel) in the same medium and incubated without shaking at 37°C for 24 hr (left hand panel) or 48 hr (right hand panel). (**D**) *E. coli* strains MCDSSAC or an isogenic *tatABC* mutant harboring pSU18 (empty vector), pSU18 producing native AmiA, or pSU18 producing either Sco3746$_{TMD}$-AmiA or PFD$_{TMD}$-AmiA fusion proteins, or variants of these where the twin-arginine motif was substituted to twin lysine (Sco3746$_{TMD}$RRKK-AmiA/PFD$_{TMD}$RRKK-AmiA) were serially diluted and spotted onto LB or LB containing 2% SDS. The plates were incubated for 20 hr at 37°C.

## Sco3746$_{TMD}$ and PFD$_{TMD}$ fusions are stably inserted in the membrane in the absence of a functional Tat system

We next determined whether these fusion proteins were stably inserted into the membrane. *Figure 8A* shows that both Sco3746$_{TMD}$-MBP and PFD$_{TMD}$-MBP were detected exclusively in the membrane fraction of a *tat*$^+$ strain at close to their theoretical masses (68 kDa for Sco3746$_{TMD}$-MBP and 81 kDa for PFD$_{TMD}$-MBP). It should be noted that the relatively poor expression of PFD$_{TMD}$-MBP necessitated long exposure times for visualisation by western blot, thus two additional non-specific bands were also detected by the MBP antibody for these samples. Substitution of the Tat consensus arginine pair for di-lysine did not detectably affect the amount of fusion proteins produced, nor their membrane localization, indicating that membrane insertion of each of these fusions occurred independently of the Tat system. This was confirmed by repeating the analysis in a *tat*$^-$ strain, where as expected the fusions were again detected exclusively in the membranes. Washing the membranes with 4 M urea or 0.2 M carbonate did not extract either protein (*Figure 8B*), indicating that they were integrally inserted into the membrane in the absence of the Tat pathway. This indicates the participation of a second protein translocase, almost certainly the Sec pathway, in the insertion of these proteins into the membrane.

## Sco3746$_{TMD}$-MBP has five TMDs

To confirm the predicted topology of the hydrophobic domain of Sco3746, we undertook a cysteine accessibility study. The Sco3746$_{TMD}$-MBP fusion is naturally devoid of cysteine residues. Guided by topology prediction programs we made three Cys substitutions (G14C, A137C and A219C) that are predicted to reside at the cytoplasmic side of the membrane and two (G84C and G171C) that are located in predicted extracellular loops (*Figure 8C*). We produced these constructs in a *tat*$^+$ strain and probed cysteine accessibility using the reagent methoxypolyethyleneglycol maleimide (MAL-PEG). This reagent, which has a mass of around 5000 Da, can pass through the outer membrane in the presence of EDTA, but is impermeable to the inner membrane. *Figure 8D* shows that the G84C and G171C variants of Sco3746$_{TMD}$-MBP clearly labelled with MAL-PEG in whole cells confirming that they are extracellular. By contrast, G14C, A137C and A219C variants were not labelled in whole cells but were labelled upon cell lysis, consistent with them having a cytoplasmic location. Taken together we conclude that the Sco3746$_{TMD}$ portion of the Sco3746$_{TMD}$-MBP fusion has 5 TMDs.

## A conserved mechanism regulates Sec-Tat transfer for dual-targeted protein families

Our prior results analysing the interaction of actinobacterial Rieske proteins with the Sec pathway indicated that a combination of low hydrophobicity of the Tat-dependent TMD coupled with the presence of positive charges close to the C-terminal end of that TMD promoted release of the polypeptide from the Sec pathway. We therefore inspected the sequences of the Sco3746 homologues, PFD proteins and YukE homologues to see whether these features are conserved across protein families. *Figure 5* shows that several non-conserved positive charges are located close to the C-terminus of TMD5 of the Sco3746 homologues examined, and analysis of predicted $\triangle G_{app}$ values for membrane insertion of the five TMDs (*Table 4*) shows a positive $\triangle G_{app}$ for TMD5 suggesting that it may potentially be poorly recognised by Sec. Interestingly, unlike the actinobacterial Rieske proteins

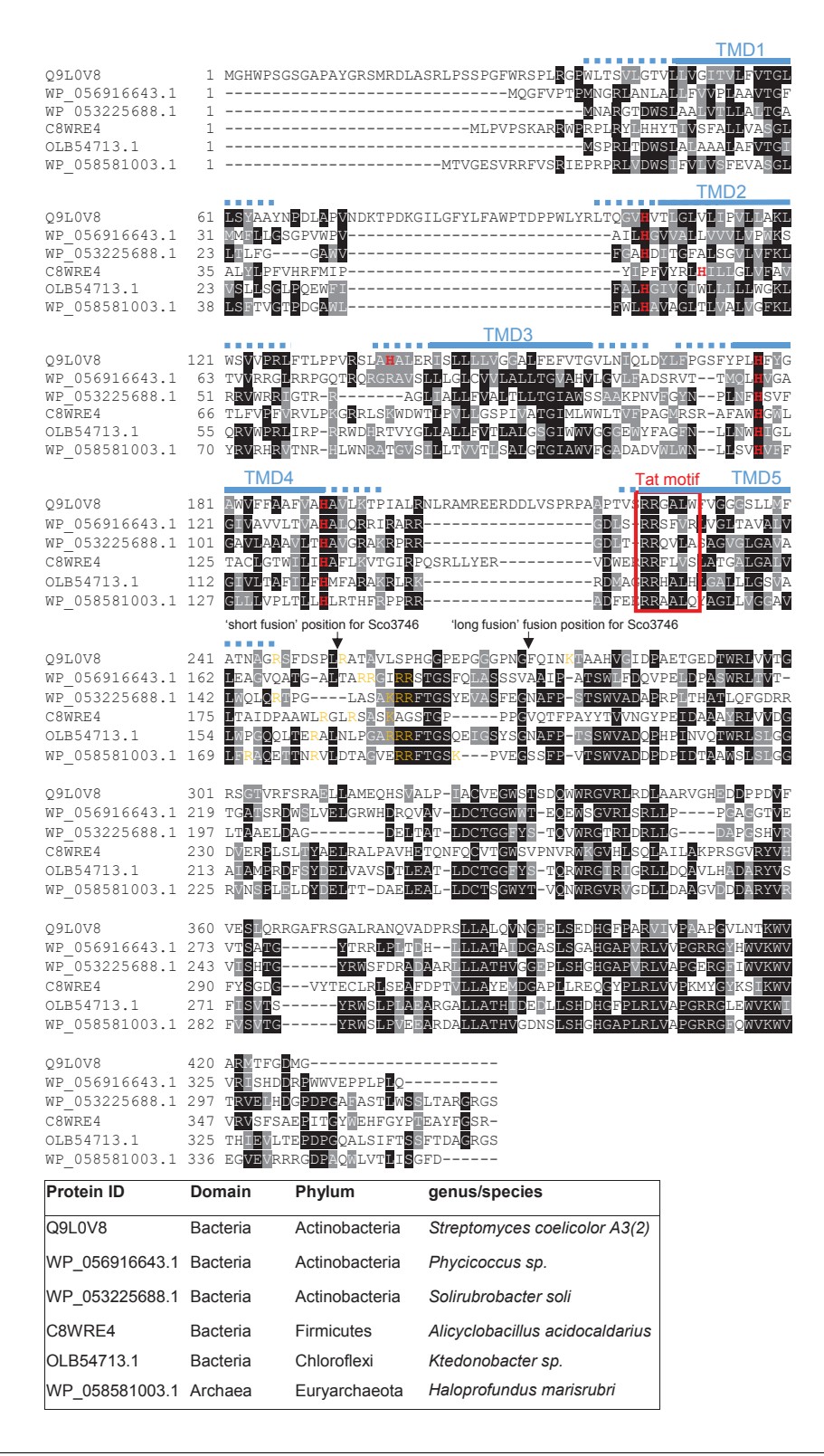

**Figure 5.** Sequence alignment of selected polytopic MoCo-binding proteins. Sequences of polytopic predicted MoCo-binding proteins from the indicated prokaryotes were aligned using ClustalW (http://www.ch.embnet.org/software/ClustalW.html) and Boxshade (http://www.ch.embnet.org/software/BOX_form.html). Predicted positions of the TMDs, using the SCAMPI2/TOPCONS servers (*Tsirigos et al., 2015*, *2016*), are shown in blue. *Figure 5 continued on next page*

Figure 5 continued

Positively charged amino acids immediately downstream of TMD5 are shown in orange. The consensus twin arginine (Tat) motif is boxed in red. Histidine residues that may co-ordinate heme *b* are shown in red font. The positions after which Bla was fused to the *S. coelicolor* protein are indicated.

which have a highly conserved loop region between Sec-dependent TMD2 and Tat-dependent TMD3, the Sco3746 homologues have non-conserved loop sequences between TMD4 and TMD5 that show apparent length variability (although all of them are predicted to be at least 8aa long,

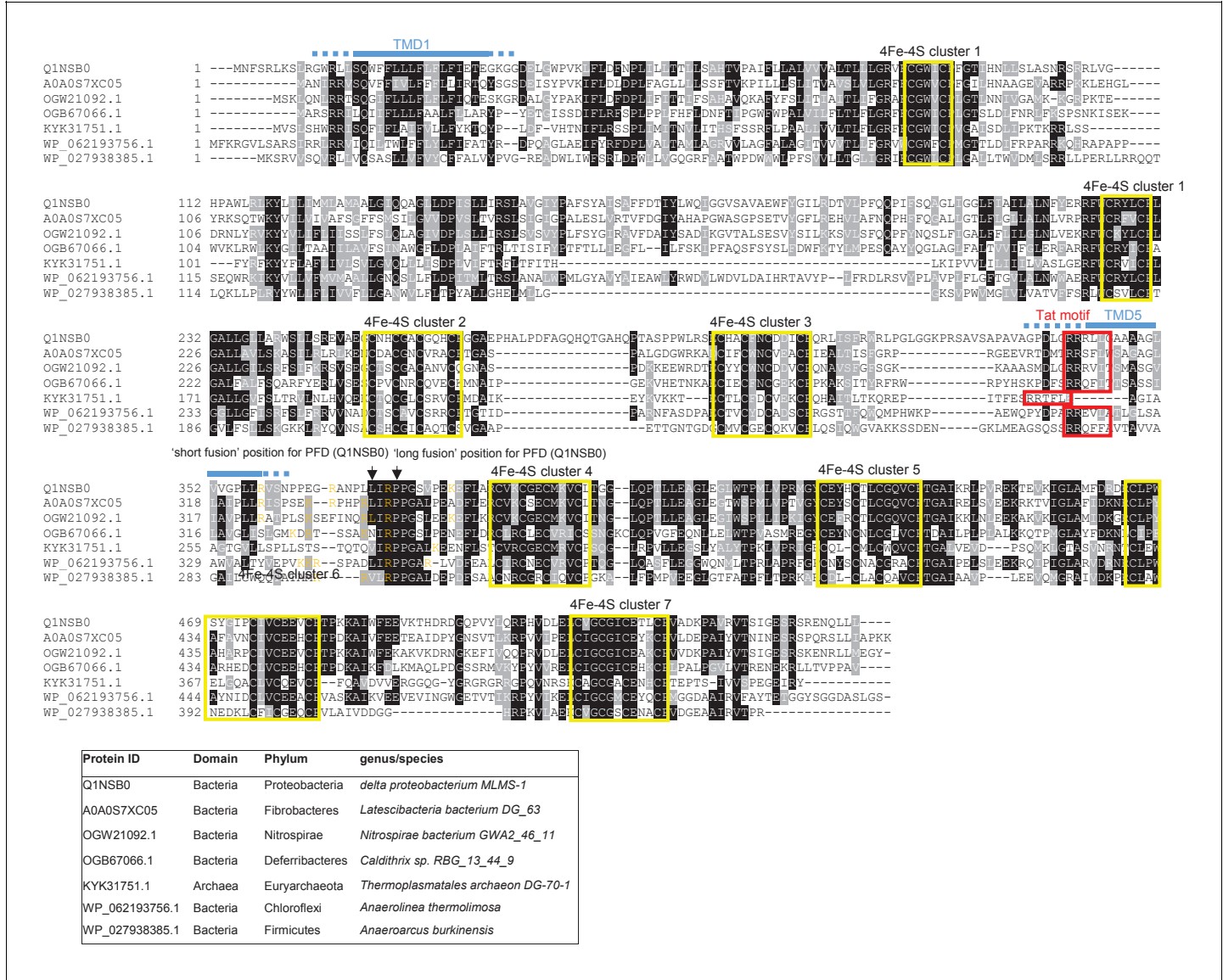

**Figure 6.** Sequence alignment of selected polytopic polyferredoxin proteins. Sequences of polytopic predicted polyferredoxin proteins from the indicated prokaryotes were aligned using ClustalW (http://www.ch.embnet.org/software/ClustalW.html) and Boxshade (http://www.ch.embnet.org/software/BOX_form.html). Predicted positions of TMD1 and 5 (using the SCAMPI2/TOPCONS servers (*Tsirigos et al., 2015*, *2016*)) are shown in blue. Positively charged amino acids immediately downstream of TMD5 are shown in orange. The consensus twin arginine motif is boxed in red and cysteine-rich regions that are predicted coordinate 4Fe-2S cluster are boxed in yellow. The positions after which Bla was fused to the delta proteobacterium *MLMS-1* protein are indicated.

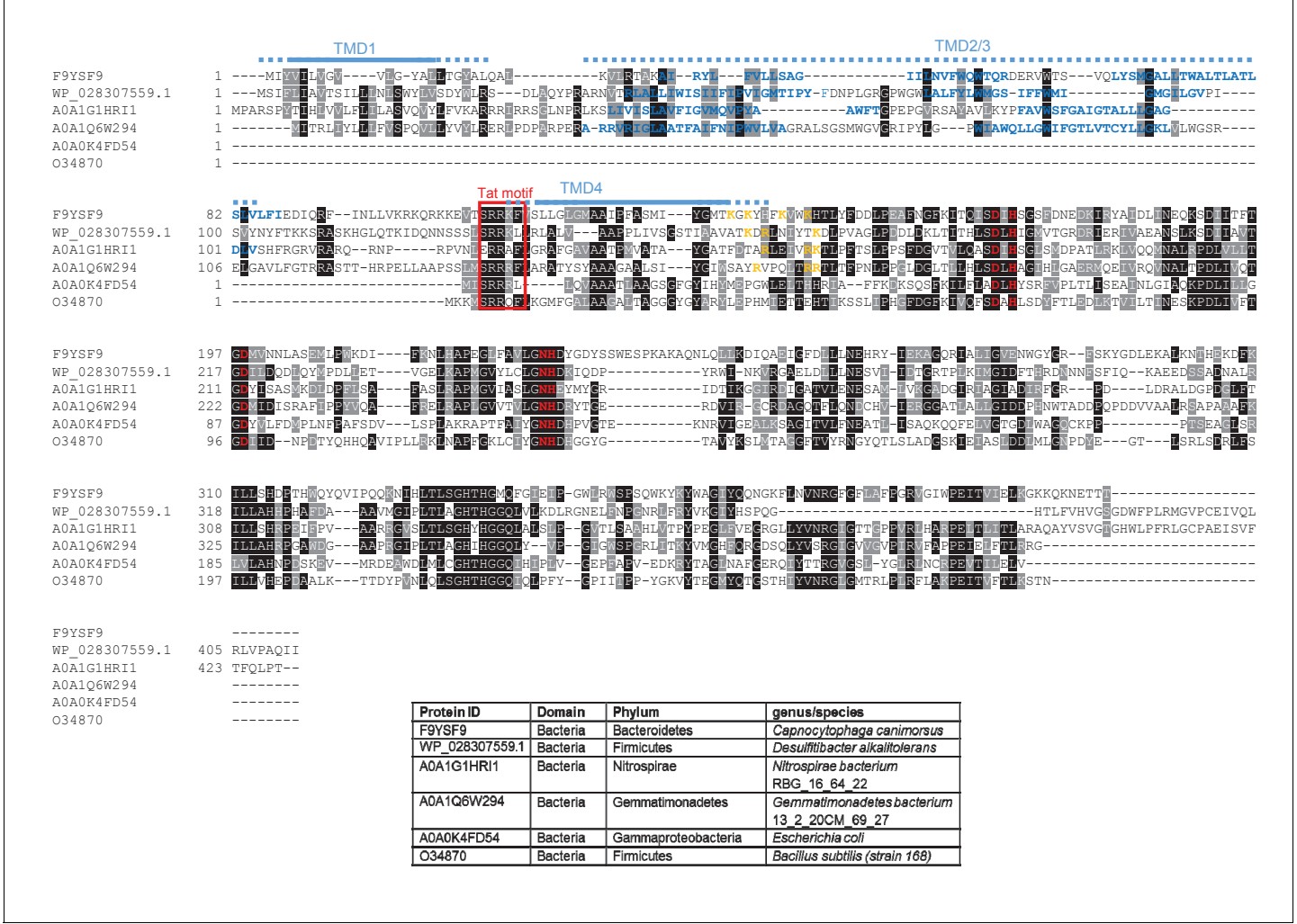

**Figure 7.** Sequence alignment of selected YkuE-related proteins. Sequences of polytopic YkuE-like metallophosphoesterase proteins from the indicated prokaryotes were aligned, alongside the shorter homologues from *E. coli* and *B. subtilis* using ClustalW (http://www.ch.embnet.org/software/ ClustalW.html) and Boxshade (http://www.ch.embnet.org/software/BOX_form.html). Predicted positions of the TMDs (using the SCAMPI2/TOPCONS servers (*Tsirigos et al., 2015*, *2016*)) are shown in blue. Note that residues in predicted TMD2 and TMD3 are not well aligned across the homologues and therefore amino acids predicted to be in TMD2 and TMD3 for each protein are individually marked in blue font. Positively charged amino acids immediately downstream of TMD5 are shown in orange. The consensus twin arginine motif is boxed in red and amino acids predicted to coordinate the metal ion cofactor are shown in red font.

which is the minimum loop length we defined for efficient recognition of Sco2149 TMD3 by the Tat pathway; *Table 2*).

We constructed 'short' (after aa 252) and 'long' (after aa 272) variants of Sco3746$_{TMD}$ fused to Bla (*Figure 9A*), and expressed these in a *tat*$^-$ strain to score for Sec-translocation of TMD5. *Figure 9B* shows that for the short fusion there is some degree of insertion of TMD5 by the Sec pathway because the M.I.C. for ampicillin mediated by this construct is significantly higher than the basal level. Substitution of hydrophobic leucines into residues towards the predicted centre of TMD5 is predicted to shift the $\triangle G_{app}$ for membrane insertion of TMD5 from positive to negative (*Table 4*), and indeed, substitution of two or more leucine residues into the short fusion doubled the M.I.C. for ampicillin (*Figure 9B*), consistent with an increased level of insertion of TMD5 by Sec. The long Sco3746$_{TMD}$-Bla fusion harbours an additional positive charge relative to the short fusion (*Figure 9A*). *Figure 9B* shows that this extension reduced the M.I.C. for ampicillin almost to the level

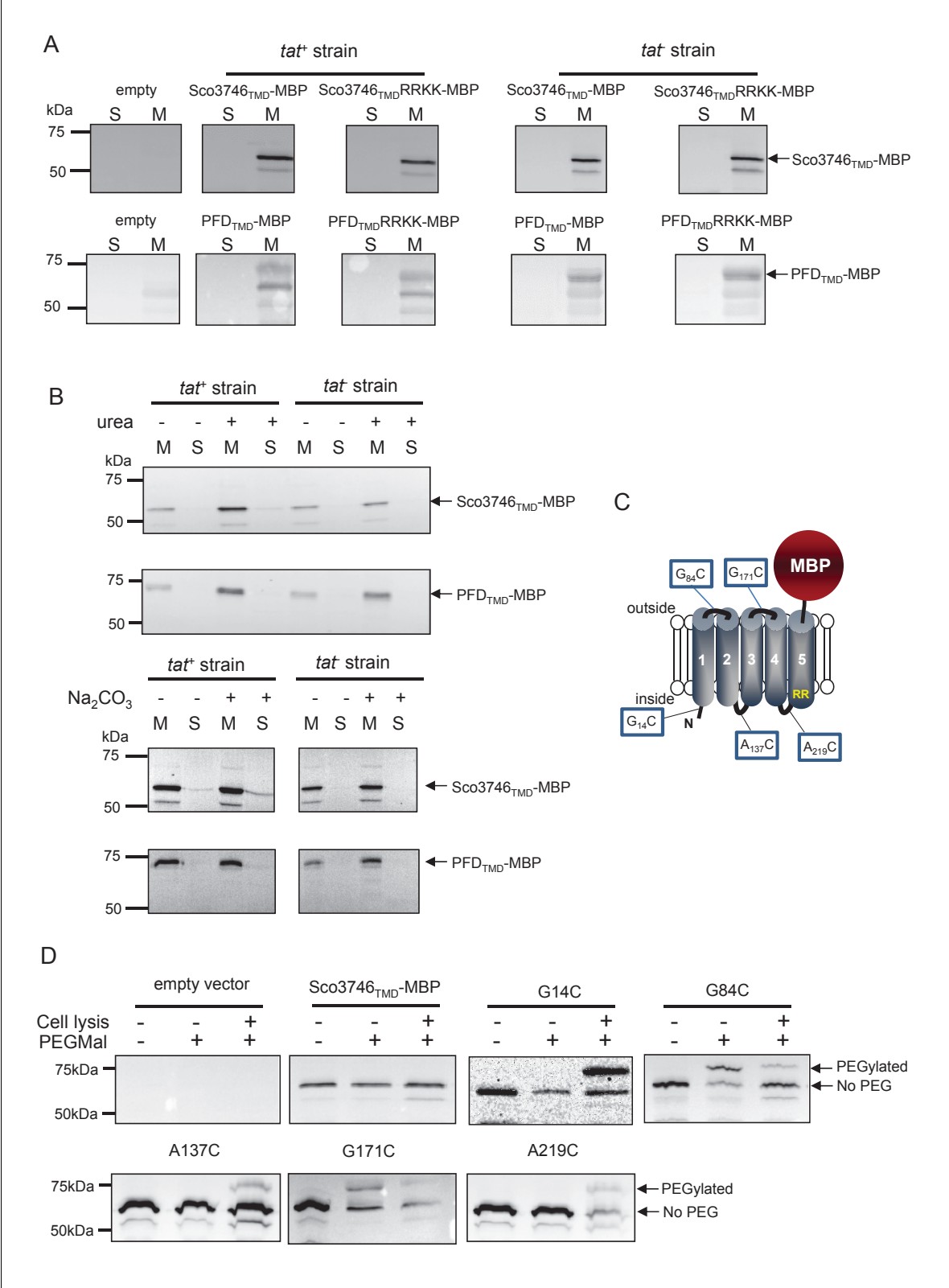

**Figure 8.** Topological analysis and membrane integration of Sco3746$_{TMD}$-MBP and PFD$_{TMD}$-MBP. (**A**) Membrane (M; 100 μg protein) and soluble (S; 50 μg protein) fractions of *E. coli* HS3018-A (△*malE*, *tat$^+$*) and HS3018-A△tat strains harboring pSU18 (empty vector), pSU18 encoding Sco3746$_{TMD}$-MBP or PFD$_{TMD}$-MBP fusion proteins, or variants of these where the twin-arginine motif was substituted to twin-lysine were separated by SDS-PAGE (12% acrylamide), transferred to nitrocellulose membrane and immunoblotted with an anti-MBP antibody. (**B**) Crude membranes of the same strains and

*Figure 8 continued on next page*

Figure 8 continued

plasmids were treated with 4M urea or 0.2M carbonate, and the presence of the fusion proteins in the wash supernatant (S) and pelleted membrane (M) was analyzed by immunoblotting as in (A). (C) Predicted locations of cysteine substitutions of Sco3746$_{TMD}$-MBP used for topology analysis. (D) Cell suspensions of strain HS3018-A harboring pSU18 alone (empty vector), or pSU18 encoding Sco3746$_{TMD}$-MBP or the indicated single cysteine substitutions of Sco3746$_{TMD}$-MBP were incubated with buffer alone, with 5 mM MAL-PEG, or were lysed by sonication and incubated with 5 mM MAL-PEG. Subsequently all samples were quenched, lysed and membranes pelleted by ultracentrifugation. Membrane samples (150 µg of protein) were separated by SDS PAGE and immunblotted as in (A).

of the empty vector control, consistent with the positive charges at the C-terminal end of Sco3746 TMD5 modulating interaction of this TMD with the Sec pathway.

Similar to Sco3746 homologues, all of the PFD proteins analysed in *Figure 6* also have a non-conserved positively charged region at the C-terminal end of TMD5. Analysis of predicted $\triangle G_{app}$ values for membrane insertion of the 5 TMDs was difficult due to variability in TMD predictions. We therefore analysed only the first and fifth (Tat-dependent) TMDs (*Table 5*), and again it can be seen that the Tat-dependent TMD has a positive predicted $\triangle G_{app}$.

To probe PFD interaction with the Sec pathway we designed 'short' (fused after aa 371) and 'long' (fused after aa 374) fusions of PFD$_{TMD}$ from delta proteobacterium *MLMS-1* to Bla (*Figure 10A*) and produced these in a *tat*$^-$ strain to score for Sec-translocation of TMD5. However, neither of these constructs mediated detectable export of β-lactamase as the M.I.C. for ampicillin was almost indistinguishable from the negative control (*Figure 10B,C*). We attribute this to the relatively poor expression of the PFD fusion proteins (e.g. *Figure 8A*). Next we substituted two, or three, leucine residues into TMD5 of PFD in each of the fusions, which is predicted to lower the $\triangle G_{app}$ value for TMD5 membrane insertion (*Table 5*). In agreement with this, *Figure 10B* shows that these substitutions significantly increased the level of interaction of the short fusion with the Sec pathway, giving mean M.I.C.s for ampicillin of 9.3 µg/ml for the G354L, R358L and 12.0 µg/ml for the G354L, P355L, R358L substitutions, respectively. These same substitutions also increased the interaction of the long fusion with Sec as they also conferred some resistance to ampicillin, each

**Table 4.** Predicted $\triangle G_{app}$ values (in kcal mol$^{-1}$) for membrane insertion of each of the five TMDs of the indicated predicted MoCo-binding proteins. Sequences were analysed using the $\triangle G_{app}$ prediction server (http://dgpred.cbr.su.se/) that are based on hydrophobicity scales generated from (*Hessa et al., 2005*, *2007*). This server uses the SCAMPI2/TOPCONS servers (*Tsirigos et al., 2015*, *2016*) to predict the positions of the TMDs and for *S. coelicolor* Q9L0V6 (Sco3746) predicts TMD1 to span aa 39–61, TMD2 to span aa 99–121, TMD3 to span aa 139–161, TMD four to span aa 173–194 and TMD5 to span aa 223–242.

| | | Predicted $\triangle G_{app}$ | | | | |
|---|---|---|---|---|---|---|
| Family/Species | Protein ID | TM1 | TM2 | Tm3[†] | Tm4[†] | TM5* |
| *Phycicoccus sp.* | WP_056916643.1 | −1.666 | −0.199 | −1.593 | 1.595 | 0.964 |
| *Solirubrobacter soli* | WP_053225688.1 | −1.778 | 1.394 | −2.239 | 1.681 | 1.135 |
| *Alicyclobacillus acidocaldarius* | C8WRE4 | −0.540 | −1.802 | −1.128 | 0.143 | 0.231 |
| *Ktedonobacter sp.* | OLB54713.1 | −0.479 | −2.647 | −2.375 | −1.257 | 1.762 |
| *Haloprofundus marisrubri* | WP_058581003.1 | 1.067 | −1.424 | −0.421 | −2.107 | 1.299 |
| *Streptomyces coelicolor* | Q9L0V6 | −2.268 | −0.011 | 0.328 | 0.512 | 1.376 |
| *S. coelicolor* G234L, S235L | | | | | | −0.757 |
| *S. coelicolor* G234L, S235L, M239L, F2440L | | | | | | −1.240 |

*Bla is fused to *S. coelicolor* Sco3746 (Q9L0V6) after amino acid 247 after amino acid 252 (full sequence of all of the fusion proteins used in this study can be found in *Supplementary file 1D*).

†Positive values for $\Delta G_{app}$ values noted for some internal TMDs. These marginally hydrophobic TMDs are, however, still likely to be integrated by the Sec pathway. Many individual TMD in multi-spanning membrane proteins have an unfavourable free energy of membrane insertion and are unable to stably integrate by themselves, requiring TMD sequence-extrinsic features for membrane insertion. It is, however, usual for the first and last TMD to be more hydrophobic as they lack these sequence-extrinsic features (*Hedin et al., 2010*; *Virkki et al., 2014*; *Elofsson and von Heijne, 2007*; *White and von Heijne, 2008b*).

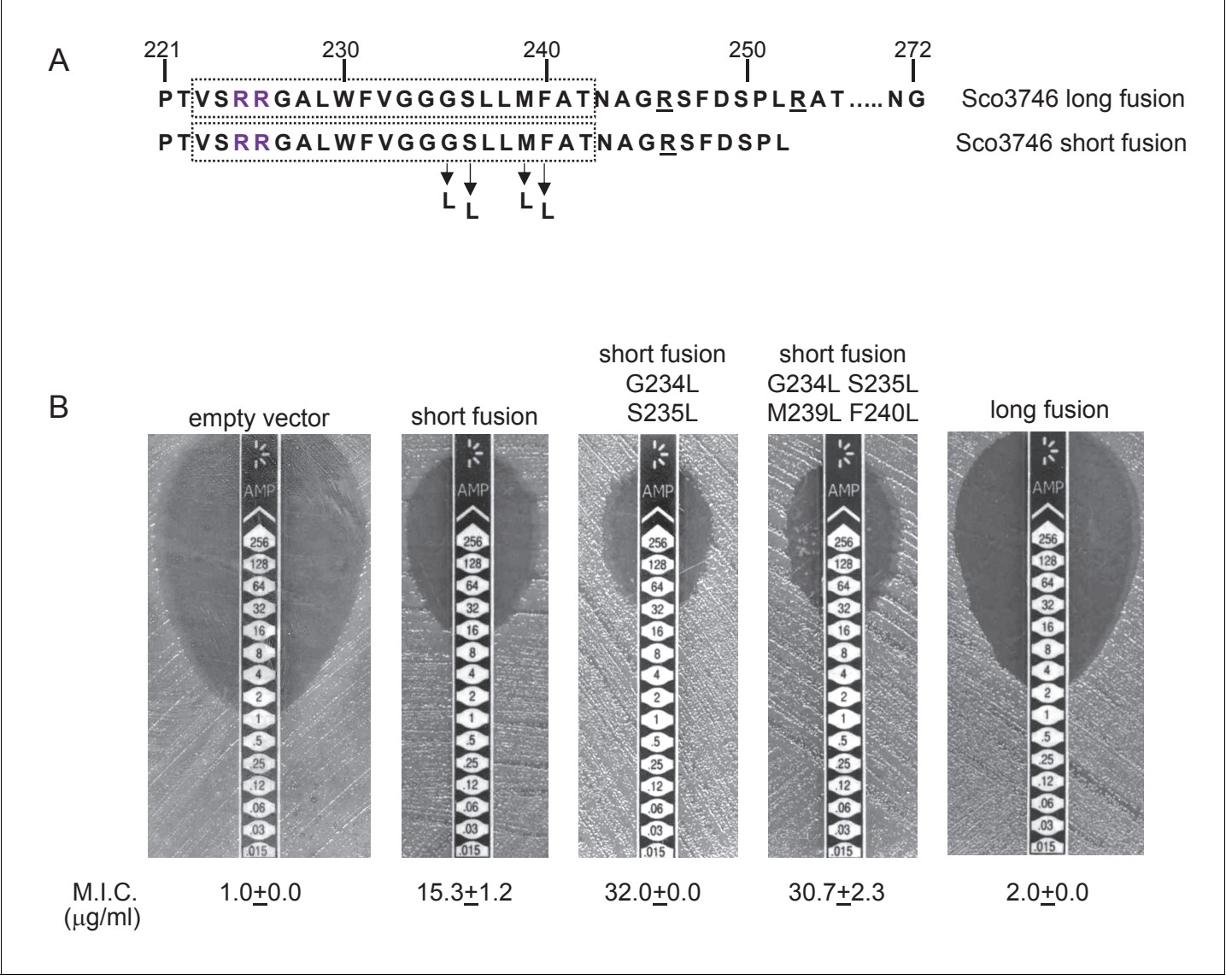

**Figure 9.** Relative hydrophobicity of TMD5 coupled with C-terminal positive charges modulate interaction of Sco3746 with the Sec pathway. (**A**) The sequence flanking TMD5 of Sco3746. The lower sequence extends to the position of the 'short' Sco3746-Bla fusion, and the amino acids in TMD5 substituted for leucine in this construct are shown. Top is the sequence fused to Bla in the 'long fusion'. The predicted position of TMD5 was determined using the SCAMPI2/TOPCONS servers (*Tsirigos et al., 2015*, *2016*) and is shown boxed. The twin arginines are shown in purple and positively charged amino acids C-terminal to TMD5 are underlined. (**B**) Representative images of M.I.C.Evaluator strip tests of strain DADE (*tat⁻*) harbouring pSU18 producing the indicated variants of Sco3746$_{TMD}$-Bla In each panel the mean M.I.C ± s.d. is given at the bottom of each test strip (where *n* = 3 biological replicates for each strain).

The following source data is available for figure 9:

**Source data 1.** Images of M. I.C.Evaluator strip tests used to generate mean M.I.C. values in *Figure 9B*.

giving a mean M.I.C. of 6.7 µg/ml (*Figure 10C*). However it is clear that the same leucine substitutions confer lower levels of resistance to ampicillin when they are present in the long construct than when they are in the short construct (compare *Figure 10B* with *Figure 10C*). Since the shorter construct harbours one less positive charge at the C-terminal end of TMD5 we conclude that the additional positive charge present in the extended fusion reduces the level of membrane insertion by Sec.

**Table 5.** Predicted $\triangle G_{app}$ values (in kcal mol$^{-1}$) for membrane insertion of the first and last TMDs of the indicated predicted polyferredoxin proteins. Sequences were analysed using the $\triangle G_{app}$ prediction server (http://dgpred.cbr.su.se/) that are based on hydrophobicity scales generated from (**Hessa et al., 2005**, **2007**). This server uses the SCAMPI2/TOPCONS servers (**Tsirigos et al., 2015**, **2016**) to predict the positions of the TMDs and for delta proteobacterium *MLMS-1* Q1NSB0 (PFD) predicts TMD1 to span aa 9–31 and TMD5 to span aa 338–359.

| Family/Species | Protein ID | Predicted $\triangle G_{app}$ | |
| --- | --- | --- | --- |
| | | TM1 | TM5* |
| *Latescibacteria bacterium DG_63* | A0A0S7XC05 | −1.792 | 0.371 |
| *Nitrospirae bacterium GWA2_46_11* | OGW21092.1 | −1.062 | 1.222 |
| *Caldithrix sp. RBG_13_44_9* | OGB67066.1 | −3.560 | 1.438 |
| *Thermoplasmatales archaeon DG-70–1* | KYK31751.1 | −1.474 | 1.715 |
| *Anaerolinea thermolimosa* | WP_062193756.1 | −2.317 | 1.107 |
| *Anaeroarcus burkinensis* | WP_027938385.1 | −0.975 | 0.836 |
| delta proteobacterium *MLMS-1* | Q1NSB0 | −2.382 | 0.297 |
| delta proteobacterium *MLMS-1* G354L, R358L | | | −0.701 |
| delta proteobacterium *MLMS-1* G354L, P355L, R358L | | | −1.741 |

*Bla is fused to delta proteobacterium *MLMS-1* PFD (Q1NSB0) after amino acid 364 (full sequence of all of the fusion proteins used in this study can be found in **Supplementary file 1D**).

Finally we noted that all of the YukE homologues we analysed in **Figure 7** also have a non-conserved positively charged region at the C-terminal end of TMD4 that is not present in the shorter, soluble variants. Moreover, analysis of predicted $\triangle G_{app}$ values for membrane insertion of the TMDs (**Table 6**) again reveals that Tat-dependent TMD has a positive predicted $\triangle G_{app}$. Taken together our results demonstrate that the mechanism of Sec release of the final TMD is conserved across three families of dual Sec-Tat targeted membrane proteins.

## Discussion

In a previous study we identified the actinobacterial Rieske FeS protein as the first protein known to be targeted to the plasma membrane by the dual action of the Sec and Tat translocases. The mechanism by which translocation is coordinated between the two pathways was not known, although a length- and sequence-conserved loop region between Sec-dependent TMD2 and Tat-dependent TMD3 was implicated in this process (**Keller et al., 2012**). Intensive investigation into the principles governing the correct biogenesis and topology of membrane proteins has revealed that the relative hydrophobicity of a TMD along with the location of positively charged amino acids are key features that govern the insertion and orientation of transmembrane segments (**Heijne, 1986**; **Hessa et al., 2005**; **Ojemalm et al., 2013**). Here we show that that these principles are exploited by nature to regulate translocation of Rieske by the Sec pathway and allow its hand-off to Tat prior to insertion of the final TMD. None of the features of the highly conserved loop region, other than the presence of one or more positively charged amino acids that serve as topology signals, plays any discernible role in co-ordinating the Sec and Tat pathways and may therefore be required for cofactor insertion or interaction with other components of the cytochrome $bc_1$ complex.

A bioinformatic analysis of prokaryotic genome sequences identified three further families of polytopic membrane proteins that share the predicted features of Sec-Tat dual-targeting. Two of these have five TMDs, with the fifth TMD immediately preceded by a consensus Tat recognition motif. A representative member of each of the 5TMD family was shown to be membrane inserted through the action of two translocases, with the Tat system recognising the final TMD. Importantly, the low hydrophobicity of the final TMD coupled with C-terminal positive charges, identified through our analysis of the *S. coelicolor* Rieske protein as being critical for Sec-release, are conserved across these further protein families, and were confirmed experimentally to govern release of this final

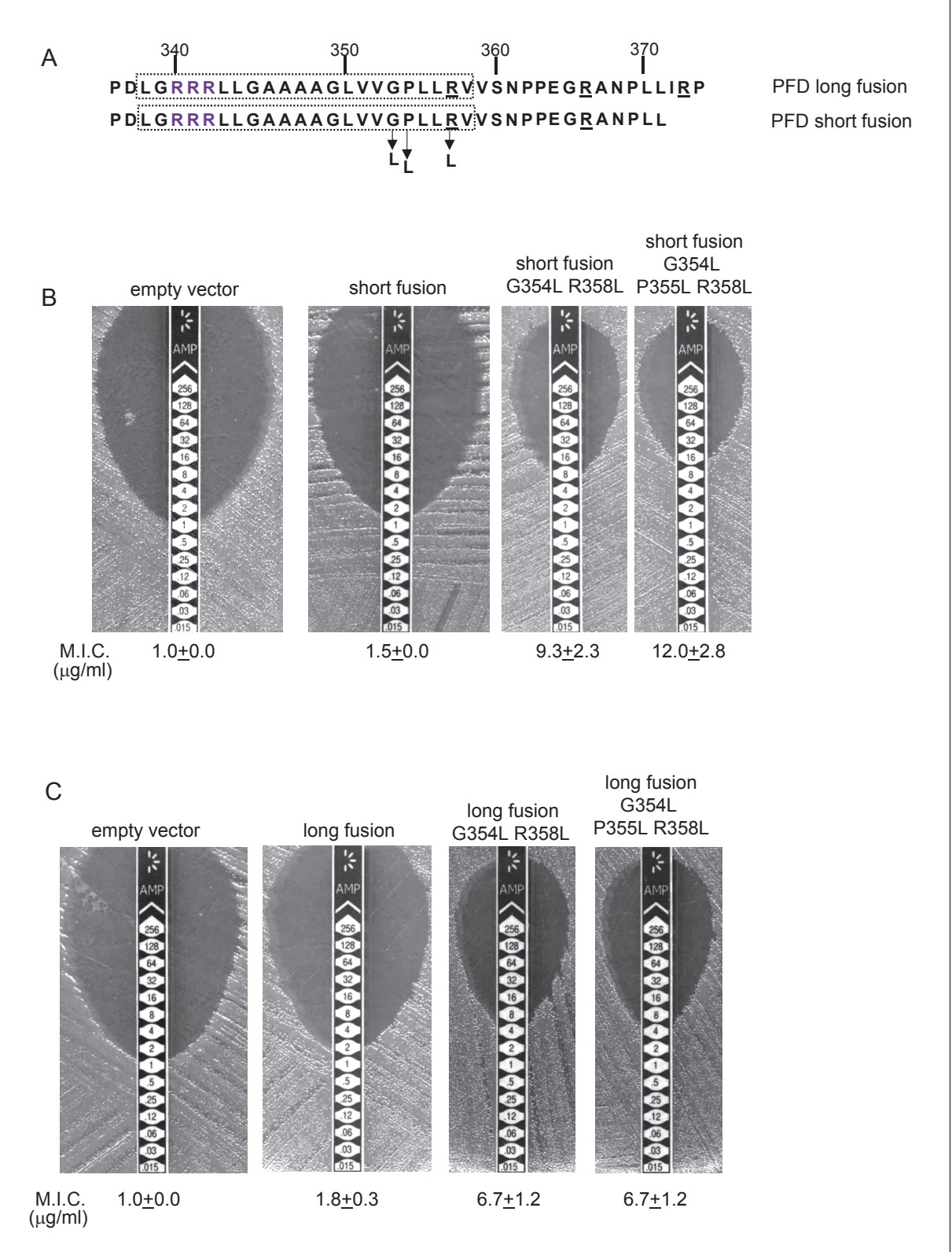

**Figure 10.** Relative hydrophobicity of *MLMS-1* PFD TMD5 coupled with C-terminal positive charges governs interaction with the Sec pathway. (**A**) The sequence flanking TMD5 of *MLMS-1* PFD. The lower sequence (corresponding to 'short fusion' in parts B and C) extends to the position of the shorter PFD-Bla fusion, whereas the top sequence is the sequence fused to Bla in the 'long fusion'. The predicted position of TMD5 was determined using the SCAMPI2/TOPCONS servers (**Tsirigos et al., 2015**, **2016**) and is shown boxed. The twin arginines are shown in purple and positively charged amino

*Figure 10 continued on next page*

*Figure 10 continued*

acids C-terminal to TMD5 are underlined. Amino acids in TMD5 substituted for leucine in both constructs are shown. (**B and C**) Representative images of M.I.C.Evaluator strip tests of strain DADE (*tat*⁻) harbouring pSU18 producing the indicated variants of PFD$_{TMD}$-Bla In each panel the mean M.I.C ± s.d. is given at the bottom of each test strip (where *n* = 3 biological replicates for each strain).

The following source data is available for figure 10:

**Source data 1.** Images of M. I.C.Evaluator strip tests used to generate mean M.I.C. values in *Figure 10B and C*.

TMD from Sec. Thus a common mechanism is at play to orchestrate the integration of dual Sec-Tat targeted membrane proteins.

A model for how such proteins are assembled is shown in *Figure 11*, using the actinobacterial Rieske protein as an example. According to the model, the Sec-dependent helices are inserted co-translationally. The positively-charged twin-arginines N-terminal to the final TMD imposes an N-in, C-out orientation on this helix. However, the C-terminal positive charges prevent the full insertion of this TMD because the relatively low hydrophobicity is insufficient to drive translocation of the C-terminal positively charged region (*Wahlberg and Spiess, 1997*; *Goder and Spiess, 2003*). This is experimentally supported by our findings that substitution of a single leucine residue into TMD3 of the *S. coelicolor* Rieske-Bla fusion is sufficient to greatly increase its Sec-dependent insertion despite the presence of two positive charges at the C-terminal end. Accordingly, it is likely that this final TMD is released by the Sec pathway as a re-entrant loop. It is formally possible that instead of polypeptide release by Sec there is direct handoff of the partially-synthesized protein to the Tat receptor complex, where its full maturation, including cofactor insertion, could potentially occur. Such a model would require interaction between the Sec and Tat machineries. However, it should be noted that the Sec and Tat pathways of *E. coli* are able to co-operatively integrate dual-targeted protein families even though the organism itself does not encode such proteins, suggesting that a direct Sec-Tat interaction is unlikely to be an essential feature of this process. Ultimately, following folding of the cofactor-containing domain, the Tat machinery mediates translocation of the folded domain across the membrane, releasing the Tat-dependent TMD into the bilayer.

Signal peptides of soluble Tat substrates often contain one or more positively-charged residues in their c-regions which are known to act as Sec-avoidance motifs. Removal of these charges results in signal sequences that can mediate efficient transport by the Sec machinery (*Bogsch et al., 1997*; *Cristóbal et al., 1999*; *Blaudeck et al., 2001*). Furthermore, signal peptides that direct proteins to the Tat machinery are known to be less hydrophobic than Sec signal peptides and if the hydrophobicity of a Tat signal peptide is increased it can also mediate efficient transport by the Sec pathway

**Table 6.** Predicted $\triangle G_{app}$ values (in kcal mol$^{-1}$) for membrane insertion of each of the four TMDs of the indicated metallophosphoesterase (YkuE) proteins. Sequences were analysed using the $\triangle G_{app}$ prediction server (http://dgpred.cbr.su.se/) that are based on hydrophobicity scales generated from (*Hessa et al., 2005*, *2007*). This server uses the SCAMPI2/TOPCONS servers (*Tsirigos et al., 2015*, *2016*) to predict the positions of the TMDs.

| Family/Species | Protein ID | Predicted $\triangle G_{app}$ | | | |
| | | TM1 | TM2 | Tm3[†] | TM4 |
|---|---|---|---|---|---|
| *Capnocytophaga canimorsus* | F9YSF9 | −1.595 | −1.327 | −1.266 | 0.957 |
| *Desulfitibacter alkalitolerans* | WP_028307559.1 | −1.409 | −2.217 | −2.075 | 1.087 |
| *Nitrospirae bacterium* RBG_16_64_22 | A0A1G1HRI1 | −1.326 | −0.017 | 0.628 | 1.579 |
| *Gemmatimonadetes bacterium* 13_2_20 CM_69_27 | A0A1Q6W294 | −1.397 | −0.016 | −0.881 | 1.282 |

[†]Positive values for $\Delta G_{app}$ value noted internal TMD. This marginally hydrophobic TMD is, however, still likely to be integrated by the Sec pathway. Many individual TMD in multi-spanning membrane proteins have an unfavourable free energy of membrane insertion and are unable to stably integrate by themselves, requiring TMD sequence-extrinsic features for membrane insertion. It is, however, usual for the first and last TMD to be more hydrophobic as they lack these sequence-extrinsic features (*Hedin et al., 2010*; *Virkki et al., 2014*; *Elofsson and von Heijne, 2007*; *White and von Heijne, 2008b*).

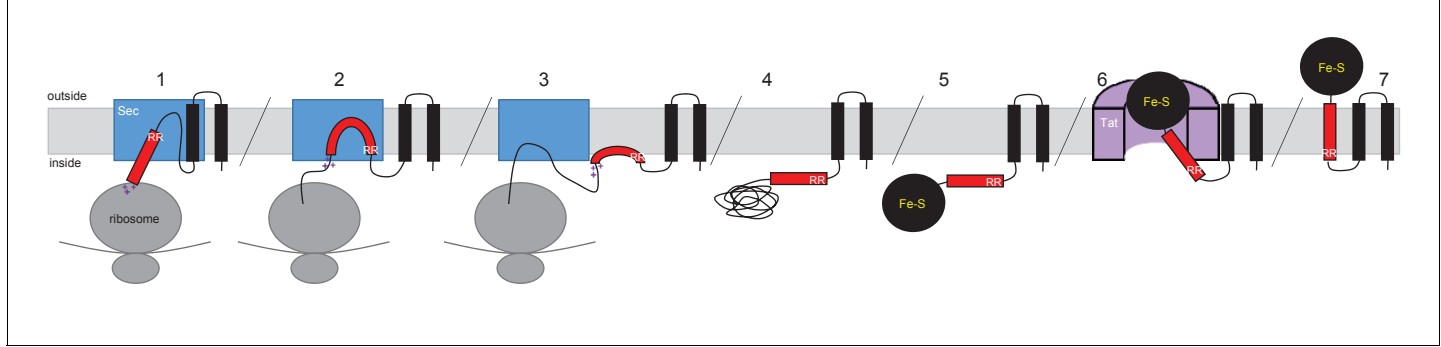

**Figure 11.** Model for actinobacterial Rieske protein assembly. 1. TMDs 1 and 2 are inserted into the membrane cotranslationally by the Sec machinery (blue box). The Sec machinery interacts with TMD3 in an N-in, C-out orientation. 2. The positive charges at the C-terminal end of TMD3 force an orientational preference on the helix and it is not inserted by the Sec machinery. 3. The hydrophobic segment of TMD3 is released from the Sec machinery as a re-entrant loop. As there are no further TMDs within the Rieske sequence the Sec machinery releases the polypeptide. 4 and 5. Translation is completed and the iron-sulfur cluster is inserted into the protein. 6. The assembled Tat machinery (pink half-cylinder) interacts with TMD3 to translocate the folded globular domain across the membrane. 7. The fully assembled Rieske protein is released into the membrane to interact with partner proteins.

(*Cristóbal et al., 1999*). However, despite possessing these 'Sec-avoidance' features, over half of the native *E. coli* Tat signal peptides are capable of transporting reporter proteins through the Sec pathway if fused to an appropriate passenger domain (*Tullman-Ercek et al., 2007*). This raises the possibility that rather than being an exception, Sec interaction with Tat signal peptides is much more frequent, and that following abortive attempts at Sec-translocation, membrane-associated twin-arginine signal peptides are common substrates of the Tat pathway. In this context it should be noted that both thylakoid and *E. coli* Tat substrates interact with the membrane before subsequent interaction with Tat machinery (*Musser and Theg, 2000*; *Ma and Cline, 2000*; *Shanmugham et al., 2006*; *Bageshwar et al., 2009*).

Our work has shown that dual targeted Sec-Tat dependent membrane proteins are dispersed across two domains including Gram-negative and Gram-positive bacteria and euryarchaea, indicating that the biogenesis of dual-targeted membrane proteins is a common feature of prokaryotes. It is interesting to note that distant homologues of both the predicted heme-Moco binding protein, Sco3746, and the *MLMS-1* polyferredoxin are widely found as separate polypeptides. For example *E. coli* MsrP/MsrQ (formerly YedY/YedZ encoded by *yedYZ*) are, respectively, a Sec-dependent polytopic heme *b* protein and Tat-targeted soluble MoCo-containing periplasmic protein that together use electrons from the respiratory chain to catalyse the repair of proteins containing methionine sulfoxide (*Brokx et al., 2005*; *Gennaris et al., 2015*). Likewise *MLMS-1* polyferredoxin is a fusion of NapH, a Sec-dependent polytopic protein with four TMD that co-ordinates [4Fe-4S] iron-sulfur clusters at the cytoplasmic side of the membrane, with NapG, a Tat-dependent periplasmic protein that is predicted to co-ordinate four [4Fe-4S] at the periplasmic side of the membrane. Collectively NapGH form a quinol dehydrogenase complex that in *E. coli* and *Wolinella succinogenes* is involved in nitrate respiration (*Brondijk et al., 2004*; *Kern and Simon, 2008*). The close relationship of such proteins and their corresponding genes raises the possibility that dual-targeted proteins arose during the course of evolution from separate polypeptides but adjacent genes. Alternatively, the ancestral proteins may have been single, dual-targeted polypeptides that subsequently separated in some organisms.

## Materials and methods

### Bacterial strains, plasmid construction and growth conditions

All strains used in this study are derived from *Escherichia coli* K-12 and are listed in *Supplementary file 1A*. Strain DH5α (Stratagene) was used for molecular biology applications. Strains MC4100 (*Casadaban and Cohen, 1979*) and DADE (as MC4100; ∆*tatABCD*, ∆*tatE*;

[*Wexler et al., 2000*]) were used for work with Bla fusions, MCDSSAC (*Ize et al., 2003*) and MCDSSACΔtat (as MCDSSAC; ΔtatABC::Apra; [*Keller et al., 2012*]) were used for work with AmiA fusions, and HS3018-A (*Caldelari et al., 2008*) and HS3018-AΔtat (As HS3018-A; ΔtatABCD, ΔtatE; [*Keller et al., 2012*]) were used for work with MBP fusions.

The amino acid sequences of all of the fusion proteins used in this study can be found in *Supplementary file 1B*. All plasmids used and generated in this study are listed in *Supplementary file 1C* and all oligonucleotides are listed in *Supplementary file 1D*. To generate pSU-PROM AmiA, DNA encoding full length AmiA was PCR amplified using oligonucleotides BamHI AmiA and SU18.2 with pSU18 AmiA (*Keller et al., 2012*) as a template, digested with *Bam*HI and *Hind*III and inserted into similarly digested pSU-PROM (*Jack et al., 2004*). To generate pSU-PROM Sco2149$_{TMD}$-AmiA, the Sco2149$_{TMD}$-AmiA allele was excised from pSU-TM123-AmiA (*Keller et al., 2012*) by digestion with *Bam*HI/*Hind*III and ligated into similarly digested pSU-PROM. To generate pSU-PROM Sco2149$_{TMD}$-Bla, the *amiA* coding region was excised from pSU-PROM Sco2149$_{TMD}$-AmiA by digestion with *Xba*I/*Hind*III and replaced with the coding sequence for the mature region of Bla obtained by PCR amplification from pBR322 that had been similarly digested. To extend the Sco2149$_{TMD}$-Bla fusion to aa205 of Sco2149, the region covering Sco2149 codons 1–205 were amplified using oligonucleotides Sco2149$_{TMD}$ and Sco2149$_{TMD}$ extension and cloned as a *Bam*HI-*Xba*I fragment into similarly digested pSU-PROM Sco2149$_{TMD}$-Bla to generate pSU-PROM Sco2149$_{TMD}$extended-Bla.

DNA encoding the first 247 amino acids of Sco3746 was PCR amplified using oligonucleotides Sco3746For and Sco3746Rev with *Streptomyces coelicolor* M145 chromosomal DNA as a template, digested with *Bgl*II and *Xba*I and inserted into pSU-PROM (*Jack et al., 2004*) that had been digested with *Bam*HI and *Xba*I. The region covering the *tat* promoter and Sco3746$_{TMD}$ coding region was excised using *Eco*RI/*Xba*I and ligated into similarly digested pSU18 (*Bartolomé et al., 1991*). Subsequently DNA encoding the mature regions of AmiA (from pSU-PROM Sco2149$_{TMD}$-AmiA) or MBP (from pTM123-MBP, (*Keller et al., 2012*) were cloned in as *Xba*I-*Hind*III fragments to give Sco3746$_{TMD}$-AmiA and Sco3746$_{TMD}$-MBP, respectively. To construct Sco3746$_{TMD}$-Bla the first 252 amino acids of Sco3746 was PCR amplified using oligonucleotides Sco3746For and Sco3746(252)Rev with *S. coelicolor* M145 chromosomal DNA as a template, digested with *Bgl*II and *Xba*I and inserted into similarly digested pSU-PROM Sco2149$_{TMD}$-Bla (thus replacing the Sco2149 coding sequence with Sco3746). Subsequently the *Xba*I site was replaced with *Kpn*I by Quickchange site-directed mutagenesis using oligonucleotides Sco3746$_{TMD}$BlaFor and Sco3746$_{TMD}$BlaRev. To extend the Sco3746$_{TMD}$-Bla fusion to aa272 of Sco3746, the region covering Sco3746 codons 1–272 were amplified using oligonucleotides SU18.1 and Sco3746$_{TMD}$extension and pSU18PROM Sco3746$_{TMD}$-Bla as template. This was digested with *Eco*RI and *Kpn*I fragment and ligated into a similarly digested pSU18PROM Sco3746$_{TMD}$-Bla as template to generate pSU18PROM Sco3746$_{TMD}$extended-Bla.

A synthetic gene encoding the transmembrane region (residues 1–227) of the Rieske protein (QcrA) from *Mycobacterium tuberculosis* strain Rv2195 was codon optimised for *E. coli* K12 expression (OPTIMIZER, [*Puigbò et al., 2007*]) and the synthetic gene was purchased ready cloned in pUC57 (GenScript). The MtbRieske$_{TMD}$ coding region was subcloned by digestion *Rca*I-*Xba*I and ligated into pBAD24 (*Guzman et al., 1995*) using vector sites *Nco*I/*Xba*I. It was then digested *Bam*HI/*Xba*I and ligated into pSU-PROM Sco2149$_{TMD}$-Bla in place of Sco2149$_{TMD}$. To extend the MtbRieske$_{TMD}$-Bla fusion to aa243 of QcrA, the region covering QcrA 1–243 were amplified using oligonucleotides MtbRieske$_{TMD}$ and MtbRieske$_{TMD}$ extension and pBAD24-QcrA as template, digested with *Bam*HI-*Xba*I and ligated into similarly digested pSU-PROM MtbRieske$_{TMD}$-Bla to generate pSU-PROM MtbRieske$_{TMD}$extended-Bla.

The transmembrane coding region (residues 1–364) of the predicted polyferredoxin (PFD) from delta proteobacterium *MLMS-1* (NCBI GI:494503356) was codon optimised for *E. coli* K12 expression (OPTIMIZER, [*Puigbò et al., 2007*]) and the synthetic gene was purchased already cloned into pBluescript (Biomatik). The PFD$_{TMD}$ coding region was excised with *Rca*I/*Xba*I and cloned into pBAD24 (*Guzman et al., 1995*) that had been digested with *Nco*I/*Xba*I. Subsequently DNA encoding the mature region of MBP (excised from pTM123-MBP [*Keller et al., 2012*]) was cloned in as an *Xba*I-*Hind*III fragment. The entire PFD$_{TMD}$-MBP coding region was subsequently excised as an *Eco*RI-*Hind*III fragment and cloned into similarly digested pSU18 (*Bartolomé et al., 1991*) to give pSU18 PFD$_{TMD}$-MBP. To construct pSU18 PFD$_{TMD}$-AmiA, the MBP coding region was excised and replaced with the AmiA coding region (as an *Xba*I/*Hind*III fragment from pSU-PROM Sco2149$_{TMD}$-AmiA). To

construct the PFD$_{TMD}$-Bla fusion (which covers up to aa371 of PFD), oligonucleotides SU18.1 and PFD$_{TMD}$BlaRev were used to amplify the PFD coding sequence (with pSU18 PFD$_{TMD}$-Bla as template). The product was digested with *Eco*RI and *Kpn*I and ligated into similarly digested pSU18-PROM Sco3746$_{TMD}$-Bla to generate PFD$_{TMD}$-Bla. The PFD$_{TMD}$ coding sequence in this construct was further extended to residue 374 using oligonucleotides SU18.1 and PFD$_{TMD}$ extension and pSU18 PFD$_{TMD}$-Bla as template. The resultant product was digested with *Eco*RI-*Kpn*I and ligated into similarly digested pSU18 PFD$_{TMD}$-Bla to generate pSU18 PFD$_{TMD}$extended-Bla.

Site-directed mutagenesis was performed using the QuickChange method (Stratagene) according to manufacturer's instructions. Deletion mutants were generated from a modified QuickChange method adapted from *Liu and Naismith (2008)*. Briefly, forward and reverse primers were designed to remove up to 5 residues at a time, overlapping by 12 nucleotides upstream and downstream of the region to be deleted with an overhang of 12 nucleotides at either end. For truncations larger than 5 residues the template used, already contained a downstream deletion of all residues but the additional 5 residues to be removed. All constructs were verified by DNA sequencing.

Unless otherwise stated, *E. coli* strains were grown aerobically overnight at 37°C in Luria-Bertani (LB) broth supplemented with appropriate antibiotic/s at the indicated final concentrations - ampicillin (125 µg/ml), kanamycin (50 µg/ml), apramycin (25 µg/ml) and chloramphenicol (25 µg/ml). Filter-sterilised SDS solution was added to the media to final concentration of 1% to 2% as indicated. Phenotypic growth tests in the presence of SDS were performed as follows: overnight cultures were diluted to OD$_{600}$ 0.1 and 5 µl aliquots were spotted in a serial dilution series from $10^4$ cells to $10^1$ cells per 5 µl for Sco2149$_{TMD}$-AmiA and $5.10^6$ to $10^5$ for Sco3746$_{TMD}$-AmiA and PFD$_{TMD}$-AmiA on LB agar supplemented with 1 or 2% SDS. Phenotypic testing for maltose fermentation employed the approach of (*Keller et al., 2012*) using maltose-bromocresol purple broth prepared with M9 minimal medium supplemented with 0.002% bromocresol purple (Roth) and 1% maltose. Growth was performed in 96-well plates incubated without shaking for 24 hr to 48 hr at 37°C. *E. coli* susceptibility to ampicillin was determined by assessing the Minimum Inhibitory Concentration (M.I.C.) that prevented growth. Stationary phase cultures were diluted to OD$_{600}$ 0.1 and LB agar plates were inoculated by swabbing the diluted culture to generate a lawn of bacteria. Oxoid M.I.C.Evaluator test strips (Thermo Fisher Scientific) containing a gradient of 0–256 µg/ml ampicillin were placed onto the lawn and incubated at 37°C for 18 hr. The M.I.C. value (in µg/ml) was read from the scale where the pointed end of the ellipse intersects the strip according to manufacturer's instructions.

Photographs of 96-well plates were captured as JPG files using a digital camera (DX AF-S NIKKOR 18–55 mm; Nikon) and colonies on agar with a digital scanner (EPSON perfection 3490 PHOTO). JPG files were imported into Gimp for cropping but otherwise were not processed.

## Subcellular fractionation

Membrane and cellular fractions were prepared as described by *Keller et al. (2012)* with modifications. *E. coli* cells were grown overnight at 37°C in LB medium with appropriate antibiotics, subcultured and harvested at OD$_{600}$ of 0.2 for cells producing Sco2149 derivatives or OD$_{600}$ of 0.5 for cells producing Sco3746 and PFD derivatives. Cells producing Sco2149 constructs were resuspended in the same volume of hypertonic buffer (20 mM Tris-HCl pH7.5/200 mM NaCl) supplemented with EDTA-free protease inhibitor (Roche). Cells producing Sco3746 or PFD constructs were diluted to give a final OD$_{600}$ of 0.2 in the same buffer. Cells were then lysed by sonication (Branson Digital Sonifier 250) and the suspension was centrifuged for 10 min at 20 000 *g* at 4°C to remove unbroken cells and large cellular debris. The resulting supernatant was then ultracentrifuged for 1 hr at 220 000 *g* at 4°C to separate membrane and soluble fractions. An aliquot of the soluble fraction was kept for analysis and the membrane pellet was resuspended in 50 mM Tris-HCl pH 7.5; 5 mM MgCl$_2$; 10% (v/v) Glycerol. Protein concentration was estimated by the Lowry method (*Lowry et al., 1951*) using the DCTM Protein Assay kit (Bio-Rad) and a standard curve generated with Bovine Serum Albumin (BSA). Membrane and soluble fractions were snap-frozen and kept at −20°C until further analysis. Urea and carbonate extraction was undertaken as described previously (*Keller et al., 2012*).

## Cysteine accessibility experiments

Cysteine accessibility was assessed as described by *Koch et al. (2012)* with the following modifications. Cultures were harvested at $OD_{600}$ of 0.8 and resuspended to give a final $OD_{600}$ of 0.3 in a labeling buffer (50 mM HEPES, 5 mM $MgCl_2$ pH 6.8) supplemented with EDTA-free protease inhibitor (Roche). An aliquot of the sample was lysed by sonication. Labeling was then performed for 1 hr at room temperature on intact or lysed cells using 5 mM methoxypolyethylene glycol maleimide (MAL-PEG) at room temperature for 1 hr in the presence of 5 mM EDTA. A control without addition of labeling reagent was systematically included. The reaction was quenched by addition of 100 mM dithiothreitol (DTT) and the whole cell samples were then lysed by sonication. All samples were subsequently centrifuged for 10 min at 20,000 *g*, 4°C to remove non-broken cells and the resulting supernatant ultracentrifuged for 30 min at 200,000 *g* at 4°C to pellet the membranes. The membrane pellet was resuspended in 60 µl of 50 mM Tris-HCl pH 7.5; 5 mM $MgCl_2$; 10% (v/v) glycerol for analysis by SDS PAGE and western blotting.

## Protein analysis

Proteins were separated by Tris-glycine SDS-PAGE (7.5%, 10%, 12% or 15% polyacrylamide, as indicated) and transferred onto nitrocellulose membrane either with semi-dry (TransBlot SD SemiDry Transfer Cell, Bio-Rad) or dry transfer (iBlot2, Life technologies). Proteins were detected with primary antibodies raised against either Sco2149 (a monoclonal Sco2149 peptide antibody generated by GenScript against the N-Terminal epitope CLPPHEPRVQDVDER), Bla (monoclonal antibody, Abcam ab12251), MBP (monoclonal antibody, NEB E8032L), BamA (polyclonal antibody, [*Lehr et al., 2010*]). Bands were revealed with chemiluminescence (Clarity Western ECL Blotting Substrate, Biorad) after incubation with secondary antibody coupled to HRP (anti-mouse IgG or Anti-Rabbit IgG, Biorad). Light-emitting bands were visualised with a CCCD camera (GeneGNOME XRQ Syngene). ImageJ (*Schneider et al., 2012*) was used for densitometry analysis. The density of Sco2149-associated signals were normalised against the BamA-associated signals which was used as a loading control. The results were expressed as percentage of the normalised signal obtained for the unmodified Sco2149$_{TMD}$-AmiA or Sco2149$_{TMD}$-Bla fusion proteins.

## Bioinformatic analysis

A Perl script was used to run all proteins from all completed prokaryotic genomes available in Genbank at the time of analysis through the TATFind program (version 1.4; [*Rose et al., 2002*]) and the TMHMM program (version 2.0c; [*Krogh et al., 2001*]). Conditions were included in the script to state that there should be an even number of TMD before the twin arginines, the number of TMD after the twin arginines should be exactly one, and the total probability of N-in should be greater than 0.9. The output of this, sorted into number of TMD prior to the twin arginines, can be found at: http://www.lifesci.dundee.ac.uk/groups/tracy_palmer/docs/CombinedTATFindTMHMMoutput.docx. Subsequently the script was updated to state that that there should be an odd number of TMD (one, three, five or seven) before the twin arginines, the number of TMD after the twin arginines should be exactly one, and the total probability of N-in should be less than 0.5. This output, sorted into number of TMD prior to the twin arginines, can be found at: http://www.lifesci.dundee.ac.uk/groups/tracy_palmer/docs/CombinedTATFindTMHMMoutput%20N-out%203.docx.

## Acknowledgements

We thank Rebecca Keller, Jeanine de Keyzer, Arnold Driessen, Gunnar von Heijne and Frank Sargent for helpful discussion and advice, Hannah Garnham for her assistance with constructing some of the cysteine substitutions used in this study and Felicity Alcock for critical reading of the manuscript.

## Additional information

### Funding

| Funder | Grant reference number | Author |
| --- | --- | --- |
| Biotechnology and Biological | BB/L000768/1 | Marion Babot |

| Sciences Research Council | | Grant Buchanan |
|---|---|---|
| Medical Research Council | MR/K500896/1 | Fiona J Tooke |
| Biotechnology and Biological Sciences Research Council | BB/J004561/1 | Govind Chandra |

The funders had no role in study design, data collection and interpretation, or the decision to submit the work for publication.

## Author contributions

FJT, MB, Formal analysis, Investigation, Methodology, Writing—original draft; GC, Formal analysis, Investigation; GB, Investigation, Methodology; TP, Conceptualization, Formal analysis, Funding acquisition, Writing—original draft

## Author ORCIDs

Tracy Palmer, http://orcid.org/0000-0001-9043-2592

# Additional files

## Supplementary files

• Supplementary file 1. Bacterial strains, plasmids and oligonucleotides used in this study, and full amino acid sequences of the fusion proteins.

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
