## [Decision Letter]

Thank you for submitting your article "A unifying mechanism for the biogenesis of membrane proteins co-operatively integrated by the Sec and Tat pathways" for consideration by *eLife*. Your article has been reviewed by three peer reviewers, one of whom, Reid Gilmore (Reviewer #1), is a member of our Board of Reviewing Editors and the evaluation has been overseen by Gisela Storz as the Senior Editor. The following individuals involved in review of your submission have agreed to reveal their identity: Steven M Theg (Reviewer #3).

The reviewers have discussed the reviews with one another and the Reviewing Editor has drafted this decision to help you prepare a revised submission.

Summary:

The manuscript from Tooke et al. examines the mechanism of integration of an interesting class of bacterial membrane proteins. These proteins contain multiple TMDs at their amino terminus that are integrated by the SEC pathway and a cofactor binding domain at the C-terminus which is translocated across the plasma membrane by the TAT pathway. The Palmer lab described the initial member of the protein class (SCO2149) in previous publications, but had not determined how the substrate was released from the SEC complex prior to integration by the TAT complex. The current manuscript does a very nice job of defining sequence features of the final TM span, and flanking regions that allow release of the protein from the SEC complex. Secondly, bioinformatic analysis led to the identification of two additional families of dual SEC/Tat substrates. Analysis of representative members of these two protein families revealed that the same features (twin arginine motif, low hydrophobicity of the final TM span, basic residues at the start of the C-terminal lumenal domain) block transport of the terminal TM span by the SEC complex and allow handoff to the TAT complex. The experimental work in this manuscript is of high quality, and the results are presented in a careful and clear manner.

Essential revisions:

1) The text describing the bioinformatics search for additional dual Sec/Tat targeted proteins stated that the protein had to have an even number of TMDs that preceding a final TM span that is preceded by the twin arginine-targeting motif. Please explain why potential dual targeted proteins have to have the N-terminus located on the cytoplasmic face of the membrane (i.e., total TMDs = 3,5, etc.). To our knowledge there is no restriction on SEC pathway integration that mandates localization of the N-terminus in the cytosol. Do the authors mean that all examples of potential dual targeted proteins that they detected (TMD>1, RR motif preceding a final TMD) had an uneven number of predicted TM span (3 in Sco2159 family, 5 in Sco3746 and PFD families)?

2) The TAT assays in Table 1 and Table 2 are not quantitative. We fully appreciate that colony dilution assays are not readily quantified. Could the authors provide additional information about the relative growth of the large deletion series that is shown in Table 2? Specifically, the table indicates that the Δ118-153 mutant grows on plates that contain SDS, while a Δ118-154 mutant does not. Does the Δ118-153 mutant show an intermediate growth rate similar to the Sco2149TMD RRKK-Ami mutant (Figure 1, Figure 1—figure supplement 2) or is it indistinguishable from the wild type construct? Perhaps the authors could add a supplemental figure showing plates of several of the large deletion constructs if the mutants like Δ118-153 do have an intermediate growth rate.

3) In Figure 7 the authors wash crude membrane fragments in 4 M urea to determine whether the MBP-substituted Sco7346 and PFD were integrated into the membrane. This may well be sufficient for this determination, but it is unusual. Membrane integration of proteins is more usually assessed as resistance to carbonate or NaOH extraction. The authors don't explain the use of urea instead of these more established techniques, nor do they include controls to show that urea was doing what they expected. Can they say a little more about why they chose this technique?

4) In Table 4 the authors report the values of ΔGapp for TM5 of different MoCo-binding proteins. When the value for TM5 is positive, they predicted that it goes in via the Tat pathway. However, Table 4 also reports positive values for ΔGapp for TM4 for four of the six proteins examined. Why are these helices (TM4) still integrated by the Sec pathway?

5) The authors repeatedly state that the hydrophobicity and flanking charges of a helix determine or mediate its release from the Sec machinery. This might be correct, but they don't directly test whether the protein exits the SEC complex before interaction with the Tat machinery. They see that a protein crosses from the Sec pathway to the Tat pathway at specific points in the targeting mechanism, but their wording (and model in Figure 10) implies that the protein is released from the Sec translocon and goes out into the membrane on its own, and encounters the Tat translocon independently. This bias ignores the possibility of a direct handoff from the Sec translocon to the Tat translocon. As this point was not addressed specifically, we would have preferred a wording that was less explicit and included both possibilities.

6) In terms of how basic residues after a signal sequence can impact protein export by the SEC pathway, the authors should consider citing the following papers: (a) an arginine added immediately after the signal peptide of the Sec-dependent β-lactamase has been shown to abolish export of β-lactamase (Summers et al., 1989, JBC 264, 20082), which is relevant because β-lactamase was used as a reporter protein in the current manuscript and (b) other studies have shown that a positive charge added after a Sec-dependent signal prevents export by the Sec translocase (Li et al., 1988, PNAS 85, 7685; Zhu et al., 1989, JBC 264, 11833; Andersson and von Heijne, 1991, PNAS 88, 9751).

---

## [Author Response]

*Essential revisions:*

*1) The text describing the bioinformatics search for additional dual Sec/Tat targeted proteins stated that the protein had to have an even number of TMDs that preceding a final TM span that is preceded by the twin arginine-targeting motif. Please explain why potential dual targeted proteins have to have the N-terminus located on the cytoplasmic face of the membrane (i.e., total TMDs = 3,5, etc.). To our knowledge there is no restriction on SEC pathway integration that mandates localization of the N-terminus in the cytosol. Do the authors mean that all examples of potential dual targeted proteins that they detected (TMD>1, RR motif preceding a final TMD) had an uneven number of predicted TM span (3 in Sco2159 family, 5 in Sco3746 and PFD families)?*

As discussed above, we have repeated our bioinformatics analysis to look for an odd number of TMD (one, three, five or seven) prior to the twin arginine motif. We have extended our Materials and methods section to explain how we did this and included a link to the new output.

*2) The TAT assays in Table 1 and Table 2 are not quantitative. We fully appreciate that colony dilution assays are not readily quantified. Could the authors provide additional information about the relative growth of the large deletion series that is shown in Table 2? Specifically, the table indicates that the Δ118-153 mutant grows on plates that contain SDS, while a Δ118-154 mutant does not. Does the Δ118-153 mutant show an intermediate growth rate similar to the Sco2149TMD RRKK-Ami mutant (Figure 1, Figure 1—figure supplement 2) or is it indistinguishable from the wild type construct? Perhaps the authors could add a supplemental figure showing plates of several of the large deletion constructs if the mutants like Δ118-153 do have an intermediate growth rate.*

There is an intermediate growth rate shown for Δ118-153 which is similar to the Sco2149TMD RRKK-AmiA mutant and slight growth shown for Δ118-154 mutant (we have adjusted to Y/N in Table 2). A supplementary figure showing all the large deletion (≥35) variants growth on SDS is now included (Figure 1—figure supplement 4).

*3) In Figure 7 the authors wash crude membrane fragments in 4 M urea to determine whether the MBP-substituted Sco7346 and PFD were integrated into the membrane. This may well be sufficient for this determination, but it is unusual. Membrane integration of proteins is more usually assessed as resistance to carbonate or NaOH extraction. The authors don't explain the use of urea instead of these more established techniques, nor do they include controls to show that urea was doing what they expected. Can they say a little more about why they chose this technique?*

Actually we assessed membrane integration using urea and also using carbonate, but only included the urea data in the original submission. We have now also included the carbonate data in Figure 7 (now re-numbered as Figure 8). We used urea because it is reported to be a commonly-used procedure to remove peripheral membrane proteins (e.g. Ohlendiek 2004. Methods Mol Biol244:283-293). We have also used it previously, for example to assess the integration of a Tat-dependent lipoprotein where it behaved similarly to carbonate extraction (Parthasarathy *et al.* 2016, J Biol Chem 291:7774–7785). In terms of controls for the urea experiment, we have good antibodies to two *E. coli* membrane proteins, TatA and TatC, but unfortunately as we are using a Δ*tatABC* strain to show that the Tat pathway is not involved in the stability of these proteins we could not use either of these. We hope that the inclusion of the carbonate extraction data is sufficient to satisfy the concerns of the reviewers.

*4) In Table 4 the authors report the values of ΔGapp for TM5 of different MoCo-binding proteins. When the value for TM5 is positive, they predicted that it goes in via the Tat pathway. However, Table 4 also reports positive values for ΔGapp for TM4 for four of the six proteins examined. Why are these helices (TM4) still integrated by the Sec pathway?*

It has been reported that 20-30% of the transmembrane helices in multi-spanning membrane proteins have an unfavourable free energy of membrane insertion (Elofsson and von Heijne, 2007, White and von Heijne,2008). These marginally hydrophobic TMs are unable to stably integrate into the membrane by themselves, they require TM sequence-extrinsic features for membrane insertion. This can include positive charges in the cytosolic flanking loops or helix-helix interactions. It is usual for the first and the final TM to be more hydrophobic as they lack these sequence-extrinsic features. Therefore, TM4 in comparison to TM5 should have some of these sequence-extrinsic features, like the positive charges located in the cytosolic loop between TM4 and TM5, to drive integration of this TM by the Sec machinery. We have now added this information as a footnote to Table 4.

*5) The authors repeatedly state that the hydrophobicity and flanking charges of a helix determine or mediate its release from the Sec machinery. This might be correct, but they don't directly test whether the protein exits the SEC complex before interaction with the Tat machinery. They see that a protein crosses from the Sec pathway to the Tat pathway at specific points in the targeting mechanism, but their wording (and model in Figure 10) implies that the protein is released from the Sec translocon and goes out into the membrane on its own, and encounters the Tat translocon independently. This bias ignores the possibility of a direct handoff from the Sec translocon to the Tat translocon. As this point was not addressed specifically, we would have preferred a wording that was less explicit and included both possibilities.*

We appreciate the reviewers’ comments here and agree that from our work we cannot rule out a direct handoff. We have now explicitly mentioned this in the Discussion. However the native proteins contain cofactors – these must be inserted into the fully synthesized protein and the domain must fold. This means that it would have to be released by the Sec translocon in order for this to happen. It is formally possible that cofactor insertion and folding could occur while the protein has been handed off to the Tat receptor complex, although there is no evidence to support this. We would also argue that we are investigating this process in *E. coli*, which encodes no dual-targeted protein families of its own. It would be hard to believe that any protein interfaces involved in a Sec-Tat interaction would be conserved in an organism that lacks native substrates of this process. We have updated our model (now Figure 11) to explicitly include the cofactor insertion/folding steps.

*6) In terms of how basic residues after a signal sequence can impact protein export by the SEC pathway, the authors should consider citing the following papers: (a) an arginine added immediately after the signal peptide of the Sec-dependent β-lactamase has been shown to abolish export of β-lactamase (Summers et al., 1989, JBC 264, 20082), which is relevant because β-lactamase was used as a reporter protein in the current manuscript and (b) other studies have shown that a positive charge added after a Sec-dependent signal prevents export by the Sec translocase (Li, Beckwith and Inouye, 1988; Zhu and Dalbey., 1989, Andersson and von Heijne, 1991).*

We thank the reviewers for pointing out these omissions. These references have been added and the numbering adjusted accordingly.